# AI-enhanced integration of genetic and medical imaging data for risk assessment of Type 2 diabetes

Yi-Jia Huang[1,2], Chun-houh Chen [2] & Hsin-Chou Yang [1,2,3,4] ✉

Type 2 diabetes (T2D) presents a formidable global health challenge, highlighted by its escalating prevalence, underscoring the critical need for precision health strategies and early detection initiatives. Leveraging artificial intelligence, particularly eXtreme Gradient Boosting (XGBoost), we devise robust risk assessment models for T2D. Drawing upon comprehensive genetic and medical imaging datasets from 68,911 individuals in the Taiwan Biobank, our models integrate Polygenic Risk Scores (PRS), Multi-image Risk Scores (MRS), and demographic variables, such as age, sex, and T2D family history. Here, we show that our model achieves an Area Under the Receiver Operating Curve (AUC) of 0.94, effectively identifying high-risk T2D subgroups. A streamlined model featuring eight key variables also maintains a high AUC of 0.939. This high accuracy for T2D risk assessment promises to catalyze early detection and preventive strategies. Moreover, we introduce an accessible online risk assessment tool for T2D, facilitating broader applicability and dissemination of our findings.

Type 2 diabetes (T2D) is a prevalent global health concern, comprising almost 90% of diabetes mellitus (DM) cases[1]. T2D is associated with severe complications such as retinopathy, nephropathy, and cardiovascular diseases, significantly impacting health and quality of life and increasing healthcare expenses[2]. Early detection and risk assessment of T2D are crucial for effective health management. T2D has a global prevalence of 6%[3]. However, in Taiwan, the prevalence is even higher, at approximately 10%. The mortality and economic burden in medical care among T2D patients increase significantly over time[4]. T2D has a polygenic and multifactorial mode of inheritance[5,6]. The significant risk factors include genetic components, food intake, and environmental exposures[7,8].

Genome-wide association studies (GWAS) have identified T2D susceptibility loci and genes, which have been used to develop T2D prediction models[9–11]. Polygenetic risk scores (PRS) and weighted PRS have attracted attention for the genetic prediction of T2D[12–14]. However, the prediction accuracy must be elevated for clinical use[15]. Recent studies have combined single nucleotide polymorphisms (SNPs) from multi-ethnic GWAS to calculate PRS and improve prediction accuracy[16,17]. Methods, such as PRS-CSx, have been developed to integrate GWAS summary statistics from multiple ethnic groups and combine multiple PRSs with weights considering linkage disequilibrium[18–20]. The use of PRS for T2D risk assessment and prediction is crucial in clinical application and precision medicine[21].

Recent smart medicine and precision health studies have highlighted the utility of medical imaging analysis in disease diagnosis and prediction, in addition to genetic markers. Moreover, previous research has demonstrated the association of several diseases with T2D[22,23], some of which can be diagnosed using medical imaging techniques. For instance, nonalcoholic fatty liver can be diagnosed through abdominal (ABD) ultrasonography[24], osteoporosis through bone mineral density (BMD)[25], and cardiovascular disease through electrocardiography (ECG)[26]. These T2D-associated diseases can be effectively diagnosed and detected using medical imaging analysis. Considering this, our study incorporates genetic markers and medical imaging data to assess the risk of T2D. This approach enables a

[1]Institute of Public Health, National Yang-Ming Chiao-Tung University, Taipei, Taiwan. [2]Institute of Statistical Science, Academia Sinica, Taipei, Taiwan. [3]Biomedical Translation Research Center, Academia Sinica, Taipei, Taiwan. [4]Department of Statistics, National Cheng Kung University, Tainan, Taiwan. ✉e-mail: hsinchou@stat.sinica.edu.tw

comprehensive evaluation and potential improvement in T2D prediction and risk assessment.

Artificial intelligence, which encompasses machine learning and deep learning, has found extensive applications in genetic research, including disease diagnosis, classification, and prediction using supervised learning[27,28]. Extreme Gradient Boosting (XGBoost), a supervised tree-based machine learning approach[29], has demonstrated superior performance in classification and prediction. Successful applications of XGBoost in precision medicine include chronic kidney disease diagnosis[30], orthopedic auxiliary classification[31], chronic obstructive pulmonary prediction[32], and multiple phenotypes prediction[33].

Taiwan Biobank (TWB), established in 2012, is a valuable resource for the integrative analysis of genetic and medical imaging data[34]. The TWB enrolled participants aged over 20 from the Han Chinese population in Taiwan and collected baseline questionnaires, blood, urine samples, and their biomarkers of lab tests, as well as genotyping data from all participants. Follow-up data, including repeated questionnaires, biomarker measurements, and medical imaging data, were collected every two to four years. Medical imaging data includes ABD, carotid artery ultrasonography (CAU), BMD, ECG, and thyroid ultrasonography (TU). The integrative analysis of genetic and medical imaging data holds great promise for disease risk assessment and prediction, as demonstrated by recent studies[35–38]. Here, we present a study integrating genome-wide SNPs and multimodality imaging data from the TWB for T2D risk assessment, marking an advancement in the field. We developed machine learning models incorporating genetic information, medical imaging, demographic variables, and other risk factors. Furthermore, we identified high-risk subgroups for T2D, providing insights into T2D precision medicine.

## Results
This study comprised two primary analyses: a genetic-centric analysis (Analysis 1; detailed in Fig. 1 and the Methods section) and a genetic-imaging integrative analysis (Analysis 2; detailed in Fig. 2 and the Methods section). Data used in the two analyses are summarized (Supplementary Table S1). A total of 68,911 participants from the TWB were included in the analysis (Fig. S1).

### Genetic-centric analysis – Comparison of prediction models
We evaluated the prediction performance under different scenarios hierarchically (the best scenario at a previous variable was given for a discussion of the next variable) in the following order: the sources and significance levels of T2D-associated SNPs (Fig. 3A and Fig. S2), T2D phenotype definitions (Fig. 3B), family history variable combinations (Fig. 3C and Fig. S3), demographic variable combinations (Fig. 3D), demographic and genetic variable combinations (Fig. 3E), and SNP and PRS combinations (Figs. 3F and 3G). The findings are summarized as follows: First, using T2D-associated SNPs from the previous large-sample-size GWAS[11] as predictors had the highest AUC of 0.557, but its AUC was not significantly higher than that used the SNPs identified by our smaller-sample-size GWAS under different thresholds of statistical significance (Fig. 3A), although our GWASs did identify some T2D-associated SNPs (Fig. S4). Second, the phenotype defined by self-reported T2D with HbA1C ≥ 6.5% or fasting glucose ≥126 mg/dL (i.e., T2D Definition IV) had the highest AUC of 0.640. Its AUC was significantly higher than the AUCs of the other three T2D definitions (Fig. 3B). Third, sibs' disease history had a significantly higher AUC of 0.732 than parents' disease history with an AUC of 0.670 ($p = 0.009$). Moreover, additive parent-and-sib disease history had the highest AUC of 0.758. Its AUC was significantly higher than parent-only ($p < 0.001$) (Fig. 3C). Fourth, a joint effect of age, sex, and additive parent-sib disease history had the highest AUC of 0.884. Its AUC was significantly higher than other demographic variable combinations, except for the combination of age and additive parent-sib disease history (Fig. 3D).

Fifth, whatever SNPs were included or not, demographic and PRS combinations outperformed the models without incorporation of PRS (Fig. 3E), although genetic factors only improved up to 3% of AUC conditional on demographic characteristics (age, sex, and family history of T2D). Finally, given T2D-associated SNPs, AUC significantly increased if PRS was included (Fig. 3F); T2D-associated SNPs provided a limited additional effect if PRS was already included (Fig. 3G).

Among different prediction models, the model with predictors PRS-CSx, age, sex, and family history of T2D had the highest AUC 0.915 (Fig. 4A) for Type VI definition of T2D based on the first testing dataset (i.e., Dataset 6' in Fig. 1). The optimal threshold, determined by the Youden index, for the fitted value that used to predict T2D or non-T2D in the XGboost model was 0.16. The Accuracy, Sensitivity, Specificity, and F1 indices were 0.843, 0.844, 0.843, and 0.672, respectively. Furthermore, the model was tested in the second independent testing dataset (i.e., Dataset 7' in Fig. 1), and a promising result similar to the first testing dataset was found: AUC = 0.905, Accuracy = 0.843, Sensitivity = 0.846, Specificity = 0.842, and F1 = 0.644. AUCs are also provided for the other three T2D definitions (Fig. S5).

The importance of each predictor was evaluated through a backward elimination procedure of variables. The optimal model incorporating age, sex, family history of T2D, and PRS achieved an AUC of 0.915. The AUC reductions upon removing individual variables are as follows: (a) Omitting the age variable resulted in an AUC of 0.839, representing a reduction of 0.076. (b) Excluding the sex variable resulted in an AUC of 0.905, with a decrease of 0.01. (c) Removing the family history of the T2D variable yielded an AUC of 0.881, with a reduction of 0.034. (d) Eliminating the PRS variable resulted in an AUC of 0.884, decreasing to 0.031. Based on the decrease in AUC, the impact size appears to be in the order of age > family history > PRS > sex. Additionally, we evaluated feature importance (see the Methods section), and the order of feature importance is family history > age > PRS > sex. Our findings consistently highlight age and family history as the most crucial risk factors for T2D.

### Genetic-centric analysis – Assessment of family history of T2D
Family history encompasses genetics and environment. We delved into the connection between the family history of T2D – treated as a graded scale (0, 1, 2, 3, and 4) – and the genetic component represented by the PRS. Through ordinal logistic regression, we observed a beta coefficient of 0.808 and an associated odds ratio (OR) of 2.24 ($p = 1.65 \times 10^{-296}$). The remarkably small p-value emphasizes the robust statistical significance, signaling a substantial association between the PRS and familial T2D status. For each incremental unit rise in an individual's PRS, their odds of belonging to a higher family history category for T2D increase by 2.24 times. This implies a tangible shift in the likelihood of different family history classifications as the PRS changes. The findings underscore a strong statistical link between genetic predisposition, as captured by the PRS, and the gradation of family history of T2D.

Furthermore, we calculated the Population Attributable Risk (PAR) by dichotomizing PRS into a high-risk group (PRS tercile >80%) and a non-high-risk group (PRS tercile <80%). Among the 59,811 participants, the breakdown was as follows: high PRS with family history ($N = 5473$), high PRS without family history ($N = 6489$), non-high PRS with family history ($N = 16,054$), and non-high PRS without family history ($N = 31,795$). The PAR estimate was 10.17%, indicating that 10.17% of the family history of T2D is attributed to genetic heritability. If considering a broader definition of the high-risk group (PRS tercile >60%) and non-high-risk group (PRS tercile <60%), the PAR estimate increased to 18.41%.

Further consideration of environmental factors, including education level, drinking experience, exercise habits, the number of exercise types, and SNP-SNP interactions with and without SNPs' main effect, did not improve T2D prediction (Supplementary Table S2).

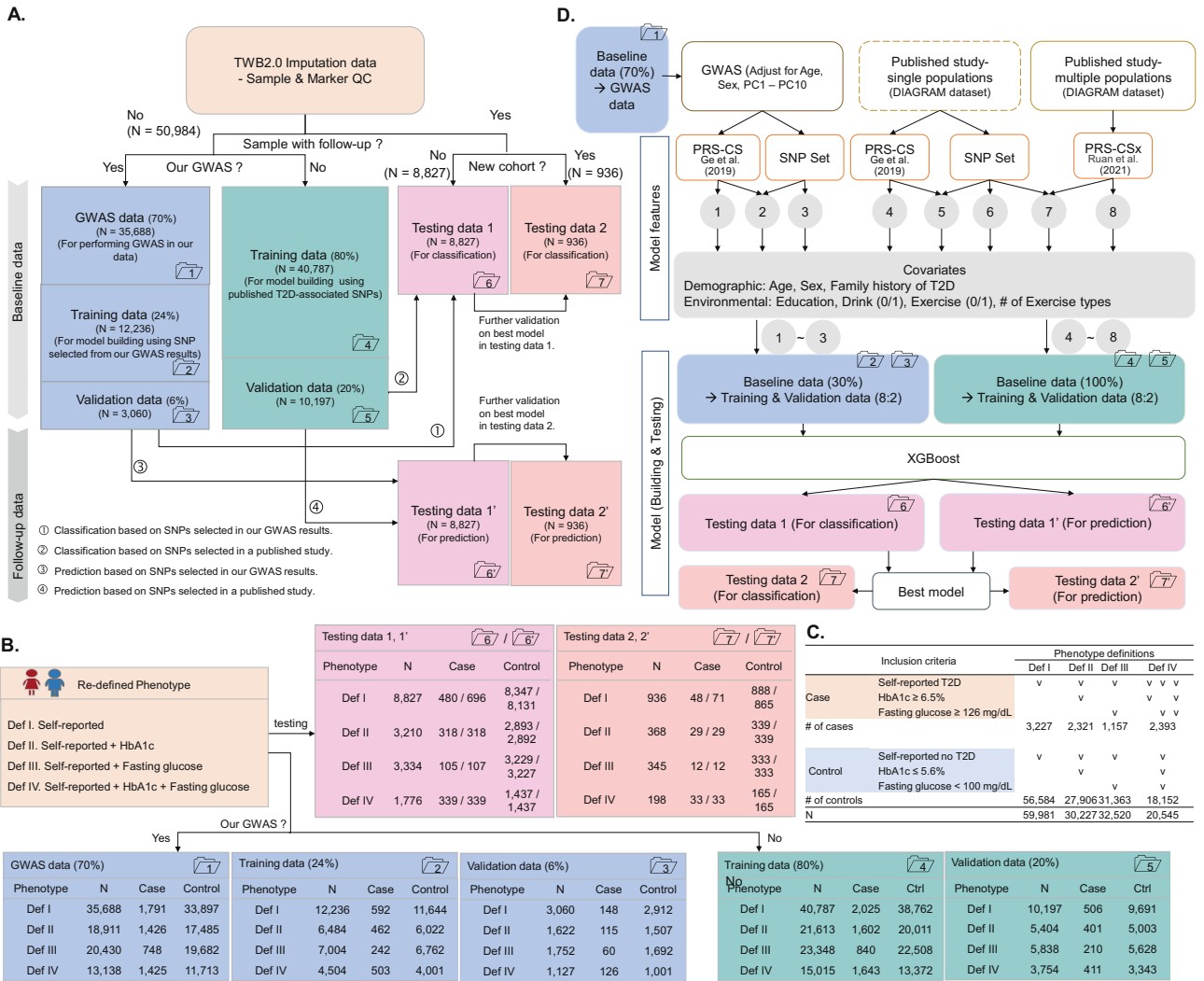

**Fig. 1 | Flowchart of genetic-centric analysis. A Data partitioning.** The dataset containing information from 60,747 individuals after data quality control (QC) was divided into several subsets: (i) The genome-wide association study (GWAS) samples (Dataset 1, *N* = 35,688), training samples (Dataset 2, *N* = 12,236; Dataset 4, *N* = 40,787), and validation samples (Dataset 3, *N* = 3060; Dataset 5, *N* = 10,197). For classification analysis, testing samples comprised Dataset 6 (*N* = 8827) and Dataset 7 (*N* = 936), while for prediction analysis, they were represented as Datasets 6' (*N* = 8827) and Dataset 7' (*N* = 936); **B Sample size**. Total sample size, along with the number of cases and the number of controls, are shown for each of the four phenotype definitions in Datasets 1 – 7; **C Phenotype definition criteria.** The

definition and sample size for the four Type 2 Diabetes (T2D) phenotype definitions is shown. **D Analysis flowchart.** The analysis flow comprises three steps, starting with selecting T2D-associated single nucleotide polymorphisms (SNPs) and polygenic risk score (PRS), then selecting demographic and environmental covariates, and the best XGBoost model was established using the selected features. As to the first step, SNPs can be chosen from **A** our own GWAS with an adjustment for age, sex, and top ten principal components (PCs), **B** published studies based on single ethnic populations, and **C** published studies based on multiple ethnic populations. Source data are provided as a Source Data file.

Considering model parsimony, the final model did not include these environmental factors and SNP-SNP interactions. In addition to prediction models, classification models were also established. The AUCs in classification models (Fig. S6) were generally slightly higher than those in prediction models (Fig. S5).

### Genetic-centric analysis – Assessment of PRS

The positive association between PRS and T2D risk is shown (Fig. 4B). Compared to the participants in the 40–60% PRS decile group, those in the top 10% decile group had a 4.738-fold risk of developing T2D (95% confidence interval: 3.147–7.132, *p* < 0.001) and a 4.660-fold risk (95% confidence interval: 2.682–8.097, *p* < 0.001) after adjusting for age, sex, and family history. In addition, we performed a stratified analysis across various combinations of age subgroups, sex subgroups, and family history subgroups to identify high-risk subgroups, where age was stratified into four subgroups based on quartiles: 0–25%, 25–50%,

50–75%, and 75–100%, corresponding to age subgroups of ≤43, 43–52, 52–59, and >59 years of age, respectively (Fig. S7). We identified a high-risk subgroup of women who were older than 59 and had a family history of T2D. The ratio of case vs. control sample size was as high as 7.3–13.0-fold in the 80–100% decile groups (Fig. 4C). The ratio was much higher than a 1.6-fold that did not consider PRS (i.e., PRS at 0–100%) (Fig. 4C). Due to ambiguity or instability in the evidence for other combinations, we chose not to report them.

### Genetic-centric analysis – Risk of developing T2D

Among 8347 non-T2D participants at baseline in the first testing dataset of 8827 participants, 220 reported T2D in the follow-up. The Cox regression analyses considered two types of time scales and three types of sex variable treatment and obtained a consistent result (Supplementary Table S3). Using the analysis in which we considered time-on-study as the time-scale with age at baseline, sex, family history

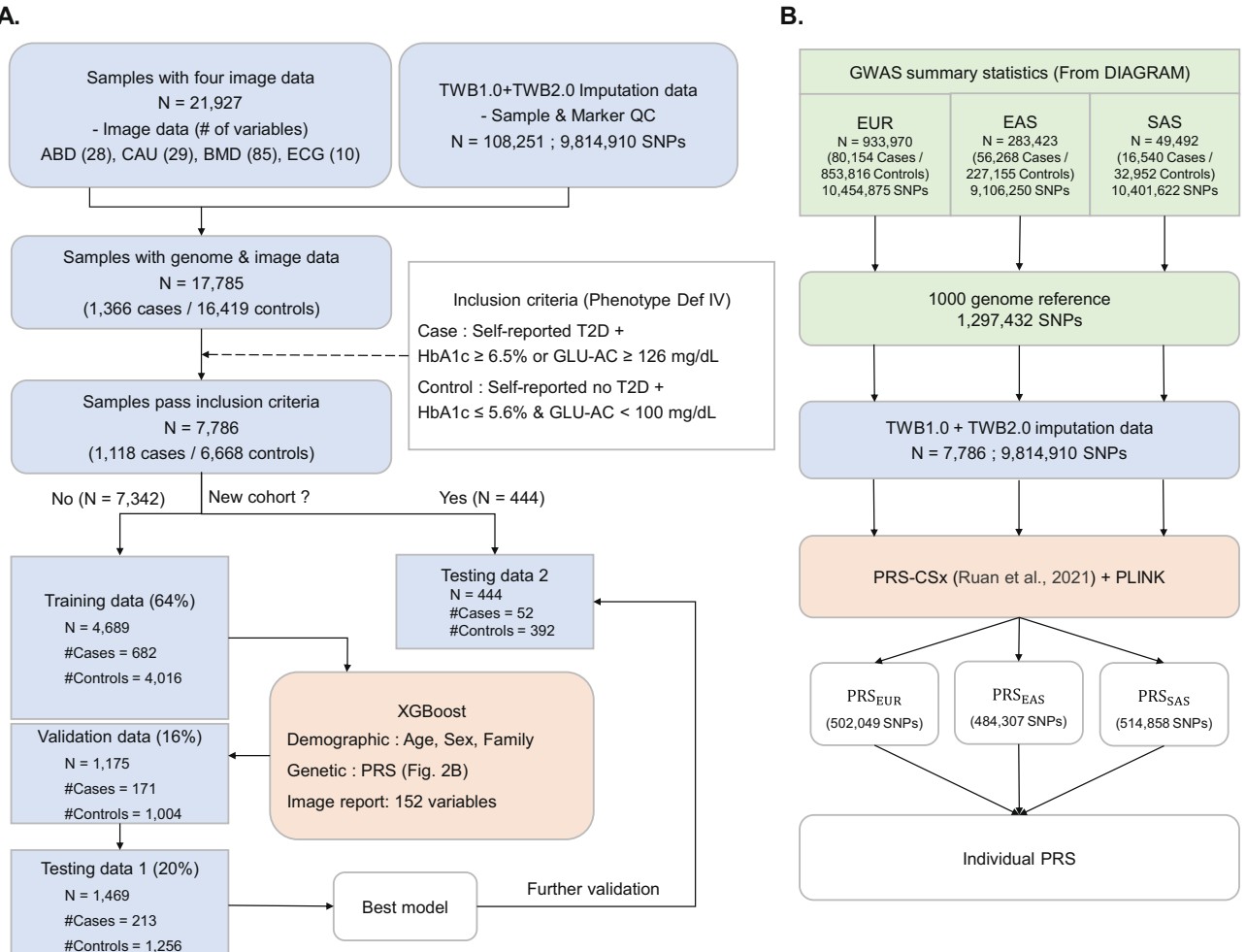

**Fig. 2 | Flowchart of genetic-image integrative analysis. A Data partitioning and model training.** Phenotype Definition IV was used as an example to illustrate the process. The data containing information from 7,786 individuals were divided into four subsets: a training dataset ($N = 4689$), a validation dataset ($N = 1175$), and two independent testing datasets ($N = 1469$ for the first dataset and $N = 444$ for the second independent dataset). Subsequently, the best XGBoost model was

established. **B Flowchart of PRS construction.** The Polygenic Risk Score (PRS) was constructed using PRS-CSx, utilizing genome-wide association study (GWAS) summary statistics from the European (EUR), East Asian (EAS), and South Asian (SAS) populations obtained from the analysis of the DIAGRAM Project. Source data are provided as a Source Data file.

of T2D, and PRS as covariates for illustration, age, sex, family history of T2D, and PRS were all significantly associated with T2D ($p < 0.001$) (Fig. 4D). Increased age, higher PRS, and stronger T2D family history had a higher T2D risk. The elderly male, with a strong family history and high PRS, had a severe T2D risk (Fig. 4E for multivariate Cox regression and Fig. S8 for univariate Cox regression). We also provided the predicted time-to-event (week) (Fig. 4F). For example, a 50-year-old man with one of his family members had T2D will achieve median T2D-free time after 460 weeks (95% CI, 384–NA). The median time to develop T2D was shortened to 419 weeks (95% CI, 384–NA) after considering a standardized PRS of 0.66 (equivalent to a PRS risk sub-group in the top 25% of the population).

A linear regression analysis was performed to assess the impact of exercise on HbA1c. Multiple testing for 110 analyses was corrected using Bonferroni correction, and the significance level was set as $4.5 \times 10^{-4}$. It was observed that individuals engaging in regular exercise experienced a significant reduction in HbA1c by an average of 0.09% mg/dL ($p < 0.001$) compared to those who did not engage in regular exercise. Moreover, individuals with a high PRS who engaged in exercise demonstrated a greater reduction in HbA1c (0.13% mg/dL) than those with a low PRS (0.08% mg/dL). The results also suggested that the T2D patients who regularly engaged in exercise can have a

noteworthy improvement of 0.32% mg/dL in HbA1c than those T2D patients who did not exercise regularly. In addition, among the various types of exercise, walking for fitness exhibited the most robust reduction in HbA1c for all samples, including high and low-risk sub-groups and both T2D and non-T2D groups (Fig. S9). On average, participants engaged in walking for fitness 18.30 times a month (standard deviation = 8.64) for approximately 48.13 minutes per session (standard deviation = 22.92).

### Genetic-centric analysis – The ability of T2D early detection in our model

To investigate the early detection capability of our model for T2D, we performed an analysis focusing on 550 women participants older than 59 years, all of whom had a family history of T2D. We identified them as at high risk if they possessed a high PRS, even though they were initially reported as non-T2D at baseline. Thirty-six were changed to T2D, and 514 were still non-T2D at follow-up. We predicted their T2D status. G1 – G4 are the groups of participants in true positive, false negative, false positive, and true negative, respectively (Fig. 5A). We evaluated that G3 was indeed misclassified by our prediction model or our prediction had corrected the problem in the self-reported T2D by further investigating: (1) their follow-up time and current risk in the Cox regression

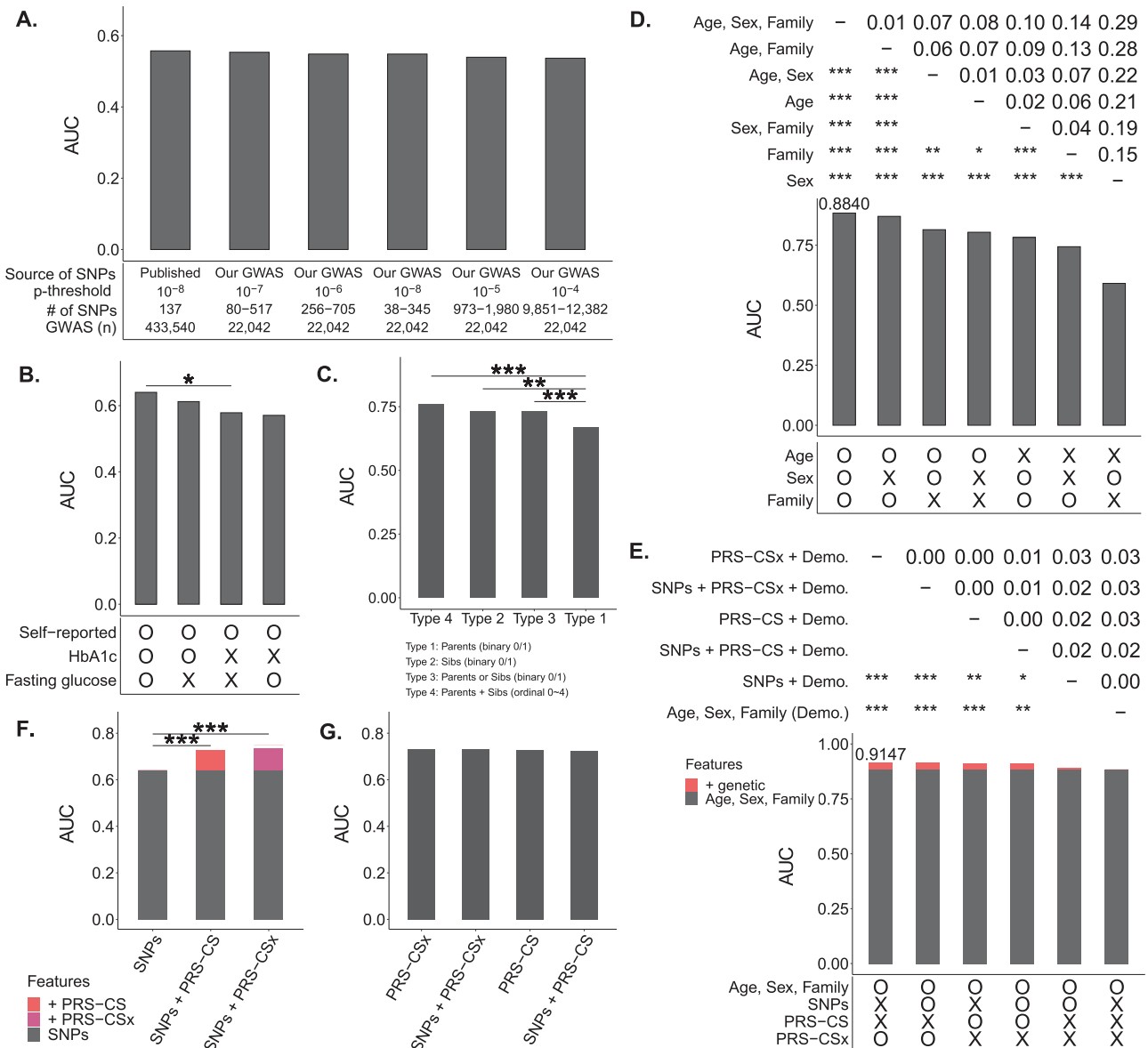

**Fig. 3 | Model evaluation and comparison.** A bar chart displays AUC. The two-sided DeLong test examined the difference between AUCs. Bonferroni's correction was applied to control for a family-wise error rate in multiple comparisons. Symbols *, **, and *** indicate p-values < 0.05, 0.01, and 0.001, respectively. **A SNP selection.** Model predictors were SNPs selected from published studies or our GWAS under different p-value thresholds, where our GWAS association test is a two-sided Wald test for the slope coefficient in a logistic regression. The average AUCs of prediction models for four phenotype definitions were compared. **B T2D Phenotype Definition.** In addition to including the selected variables in Fig. 3A, the AUCs of four phenotype definitions were compared. **C Family history of T2D.** In addition to including the selected variables in Fig. 3A, B, the AUCs of the four types of T2D family history (i.e., (i): parents (binary factor), (ii) sibs (binary factor), (iii) either parents or sibs (binary factors), and (iv) both parents and sibs (ordinal factor)) were compared. **D Demographic variables.** In addition to including the selected variables in Fig. 3A–C, the AUCs of different combinations of demographic factors, including age, sex, and family history of T2D, are compared. **E PRS and demographic variables.** In addition to including the selected variables in Fig. 3A–D, the AUCs of different combinations of genetic variables, including SNPs, PRS-CS, and PRS-CSx, and demographic variables, including age, sex, and family history of T2D, are compared. **F Impact of including PRS after SNPs.** The AUCs of the models that consider SNPs, SNPs+PRS-CS, and SNPs+PRS-CSx as predictors are compared. **G Impact of including additional SNPs after PRS.** The additional 137 SNPs were collected from published studies (Supplemental Text 2). The AUCs of the models that consider additional SNPs given PRS in the model are compared. Source data are provided as a Source Data file.

model; (2) HbA1c and fasting glucose; (3) the accuracy of self-reported disease status.

The Kaplan-Meier curve for each subgroup is depicted (Fig. 5B). The distributions of median survival time for each subgroup are illustrated (Fig. 5C). The distributions of the time period from baseline to follow-up for each subgroup are presented (Fig. 5D). The distributions of Type 2 diabetes (T2D) risk at follow-up for each subgroup are shown (Fig. 5E). The distributions of HbA1c levels at baseline and follow-up for each subgroup are displayed (Fig. 5F). The

distributions of fasting glucose levels at baseline and follow-up are demonstrated (Fig. 5G).

First, compared to G4 (true negative), G3 had a significantly lower T2D-free probability (Fig. 5B), shorter median survival time (Fig. 5C), higher T2D-risk under similar follow-up time (Fig. 5D and 5E), higher HbA1c (Fig. 5F), and higher fasting glucose (Fig. 5G). Second, compared to G1 (true positive), G3 had a comparable survival rate (Fig. 5B), median survival time (Fig. 5C), and T2D-risk under similar follow-up time (Fig. 5D and 5E) but lower HbA1c (Fig. 5F) and fasting glucose

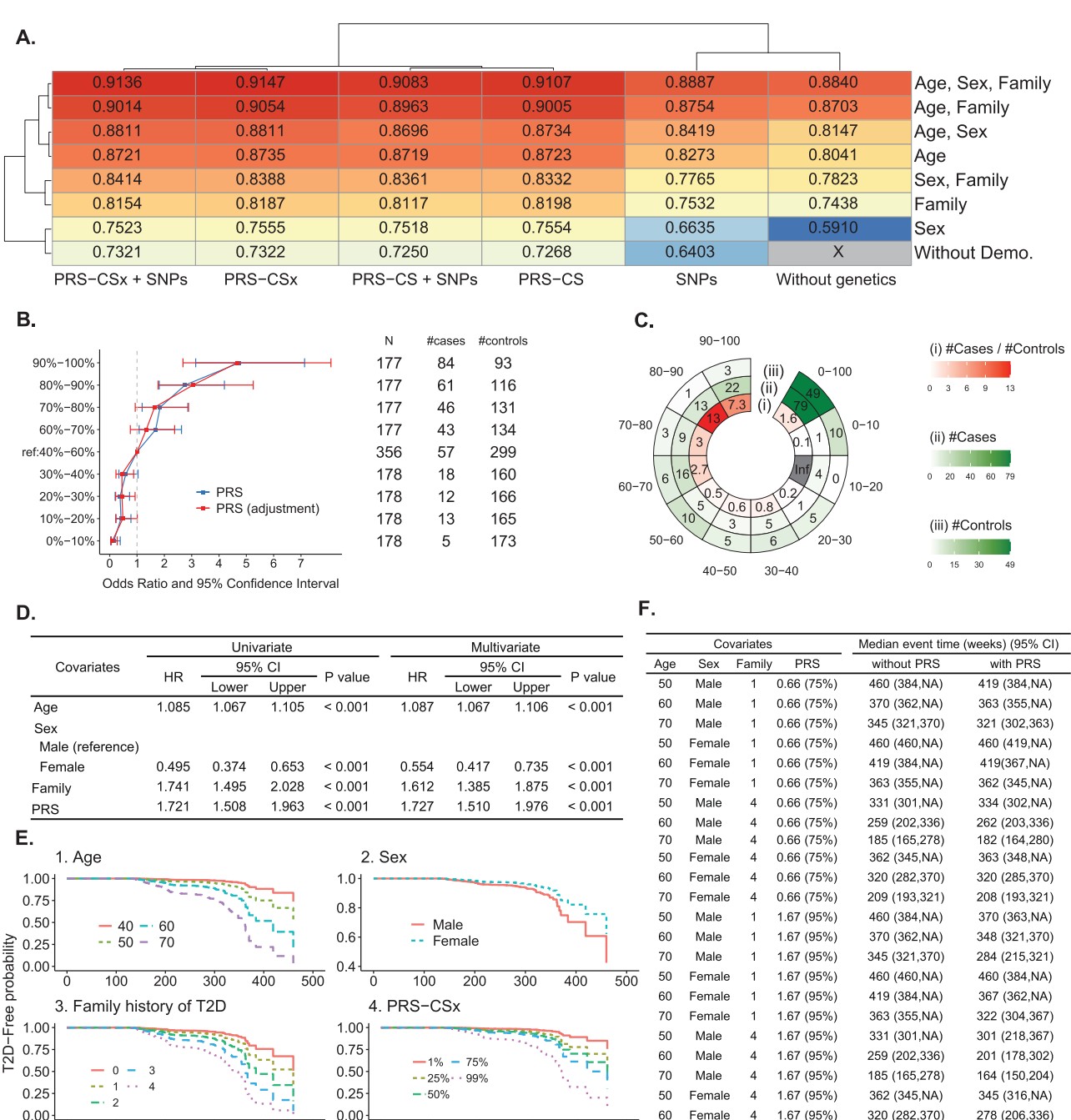

**Fig. 4 | Results in the genetic-centric analysis. A AUCs of all models based on Phenotype Definition IV.** A heatmap summarizes the AUCs of all models based on Phenotype Definition IV (i.e., T2D was defined by self-reported T2D, HbA1c, and fasting glucose). The genetic variables are shown on the X-axis, and the demographic variables are shown on the Y-axis. **B Positive correlation between PRS and T2D odds ratio.** In each decile of PRS based on PRS-CSx, the odds ratio of T2D risk and its 95% confidence interval were calculated based on an unadjusted model (blue line) and an adjusted model considering age, sex, and T2D family history (red line). The reference group was the PRS group in the 40–60% decile. The horizontal bars are presented as the odds ratio estimates (square symbol) +/− its 95% confidence intervals (left and right ends) at a PRS decile. **C High-risk group.** In the chart, the figures from the inner to the outer represent (i) the case-to-control ratio, (ii) the number of cases, and (iii) the number of controls in the PRS decile sub-groups. **D Association of age, sex, T2D family history, and PRS with T2D.** In the univariate analysis, the p-values for age, sex, family history, and PRS were $4.17 \times 10^{-20}$, $7.08 \times 10^{-7}$, $9.41 \times 10^{-13}$, and $2.06 \times 10^{-13}$, respectively. In the multivariate analysis, the p-values for age, sex, family history, and PRS were $2.00 \times 10^{-16}$, $5.56 \times 10^{-5}$, $1.43 \times 10^{-10}$, and $5.49 \times 10^{-13}$, respectively. **E Risk factors for T2D.** Kaplan-Meier curves reveal that Age (older individuals), sex (males), T2D family history (the larger number of parents and siblings who had T2D), and PRS (high decile PRS group) are risk factors (high-risk level) for T2D risk. **F Median event time of T2D.** Examples of the median event time for developing T2D are provided based on a multivariate Cox regression model, both without and with incorporating PRS. NA indicates not assessable. Source data are provided as a Source Data file.

(Fig. 5G). We didn't compare G2 and G3 because of the small sample size in G2. Finally, among the 395 participants in G3, 80.76% of them were removed from our previous analysis because their baseline HbA1c and fasting glucose violated the criteria for the phenotype definition (Fig. 1C); 339 participants were removed because of their follow-up HbA1c and fasting glucose violated the formal T2D criteria; only 34 self-reported non-T2D were really non-T2D participants who had HbA1C < 6.5% and fasting glucose <126 mg/dL (Fig. 5H). Overall,

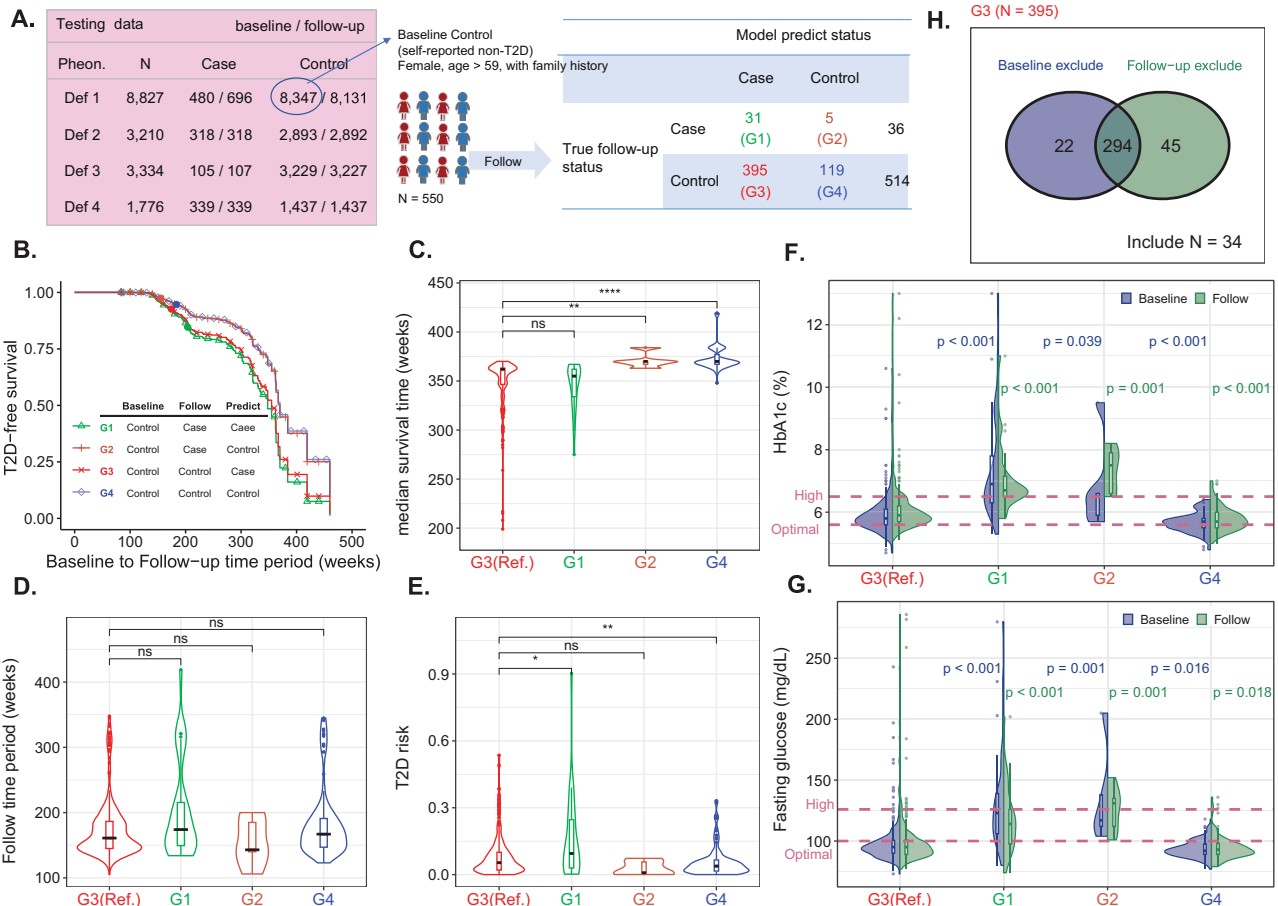

**Fig. 5 | T2D early detection using our prediction model (Phenotype Definition IV; age, sex, family, and PRS). A** Four subgroups (N = 550). **B** Survival rate (N = 550). **C** Median survival time (N = 550). P-values of G1 vs. G3, G2 vs. G3, and G4 vs. G3 were 0.092, 0.0014 (**), and 2.22 × 10⁻¹⁶ (***), respectively. **D Follow-up time** (N = 550). P-values of G1 vs. G3, G2 vs. G3, and G4 vs. G3 were 0.056, 0.32, and 0.14, respectively. **E T2D risk** (N = 550). P-values of G1 vs. G3, G2 vs. G3, and G4 vs. G3 were 0.018 (*), 0.073, and 0.0039 (**), respectively. **F HbA1c** (N = 550). P-values of G1 vs. G3, G2 vs. G3, and G4 vs. G3 were 2.21 × 10⁻¹⁴, 0.0039, and 3.00 × 10⁻⁵; respectively; in the follow-up, p-values of G1 vs. G3, G2 vs. G3, and G4 vs. G3 were 1.50 × 10⁻¹³, 6.01 × 10⁻⁴, and 4.55 × 10⁻⁶, respectively. **G Fasting glucose** (N = 550). In

the baseline, p-values of G1 vs. G3, G2 vs. G3, and G4 vs. G3 were 2.06 × 10⁻¹², 6.66 × 10⁻⁴, and 1.63 × 10⁻²; respectively; in the follow-up, p-values of G1 vs. G3, G2 vs. G3, and G4 vs. G3 were 8.30 × 10⁻⁸, 1.38 × 10⁻³, and 1.84 × 10⁻², respectively. **H Phenotype definition in G3** (N = 395). Many individuals in G3 cannot satisfy the T2D Phenotype Definition IV. In Fig. 5C–G, two-sided Wilcoxon rank-sum tests were applied to compare group differences. The box plots' center lines indicate the medians, the lower and upper boundaries of the boxes represent the first and third quartiles, and the whiskers extend to cover a range of 1.5 interquartile distances from the edges. The violin plots' upper and lower bounds depict the minimum and maximum values. Source data are provided as a Source Data file.

the results consistently indicate that G3 represents individuals in a pre-T2D stage, which our model can detect early.

**Genetic-imaging integrative analysis – Model performance and essential features**

The model that combined four types of image features performed best. Moreover, the model based on BMD image features exhibited a higher AUC, accuracy, specificity, and F1 than the models based on any other three types of images (Fig. 6A). The models based on image features had an AUC of 0.898, higher than the ones of genetic information (AUC = 0.677) and demographic factors (AUC = 0.843). Integrating image features, genetic information, and demographic factors increased AUC to 0.949 in the first testing data (Fig. 6B); the results for each of the four images are also provided (Fig. S10). The accuracy, sensitivity, specificity, and F1 of the model in the first testing data were 0.871, 0.878, 0.870, and 0.663, respectively, based on a classification threshold of 0.03. The model also performed reasonably well in the second testing dataset with AUC = 0.929, Accuracy = 0.854, Sensitivity = 0.789, Specificity = 0.862, and F1 = 0.558. The results of a prediction model using tuned parameters are also provided (Supplementary Table S4). As no significant improvement was observed, this paper

discusses the default model. According to the estimated feature importance in the best XGBoost model, all genetic factors (PRS), four types of medical images, and demographic variables provided informative features for risk assessment, such as PRS (genetics), family history and age (demographic factors), fatty liver (ABD images), end-diastolic velocity in the right common carotid artery (CAU images), RR interval (ECG images), and spine thickness (BMD images). Of the 152 medical imaging features, 125 were selected in the final model. (Fig. 6C).

To address the challenges of practical clinical implementation in the best XGBoost model, we have proposed an alternative model that requires a limited number of features. We systematically calculated each feature's incremental area under the AUC by sequentially including those with the highest feature importance. We selected the top features showing a positive AUC increment. The analysis revealed that a sub-model incorporating only the following eight crucial variables: family history (from the questionnaire), age (from the questionnaire), fatty liver (from ABD images), spine thickness (from BMD images), PRS (from genetic data), end-diastolic velocity in the right common carotid artery (R_CCA_EDV) (from CAU images), RR interval (from ECG images), and end-diastolic velocity in the left

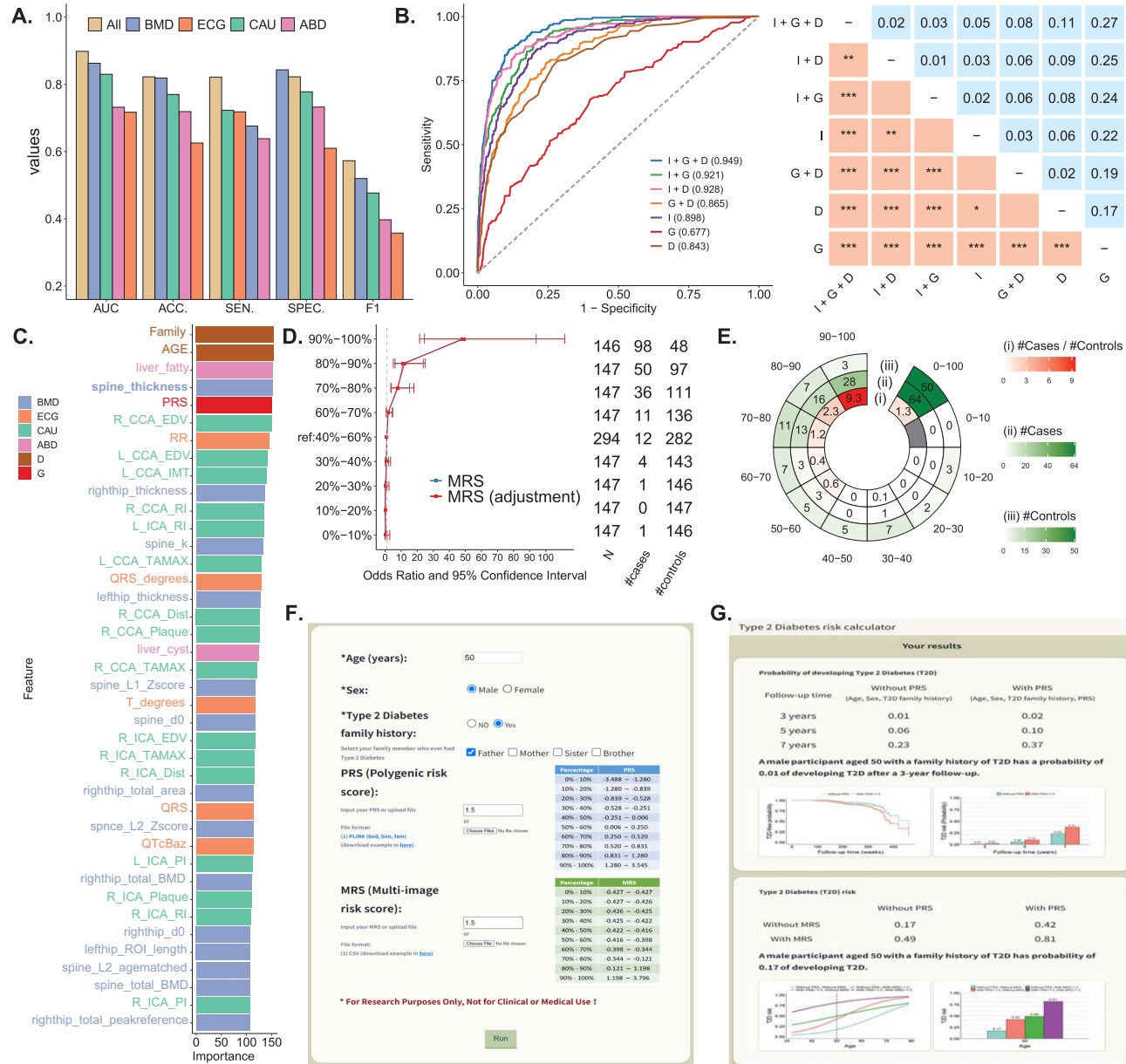

**Fig. 6 | Results in the genetic-image integrative analysis. A Performance comparison of medical imaging data analysis.** The area under the receiver operating characteristic (ROC) curve (AUC), accuracy (ACC), sensitivity (SEN), specificity (SPEC), and F1 score are compared for the integrative analysis of four types of medical images (All) and individual medical image analyses, including BMD, ECG, CAU, and ABD. **B The model that combines four types of medical imaging, PRS, and demographic variables shows the highest AUC of 0.949.** ROC plots and the corresponding AUC for the models considering medical image features (I), genetic PRS (G), and demographic variables, including age, sex, T2D family history (D), and their combinations. **C An optimal model combining medical imaging, PRS, and demographic variables.** The best model's top 20 features with a high feature impact include the medical image, genetic, and demographic features. **D Positive correlation between MRS and T2D odds ratio.** In each decile of MRS based on

four types of medical images, the odds ratio of T2D risk and its 95% confidence interval were calculated based on an unadjusted model (blue line) and an adjusted model considering age, sex, and T2D family history (red line), with the MRS group in the 40–60% decile serving as the reference group. The horizontal bars are presented as the odds ratio estimates (square symbol) +/− its 95% confidence intervals (left and right ends). **E High-risk group.** The figures from the inner to the outer in the chart display (i) the case-to-control ratio, (ii) the number of cases, and (iii) the number of controls in the MRS decile subgroups. **F Input page of the online T2D prediction website.** Personal information, including age, sex, family history of T2D, PRS, and MRS, is input to calculate T2D risk. PRS and MRS are optional, and a reference distribution is provided. **G Output page of the online T2D prediction website.** Source data are provided as a Source Data file.

common carotid artery (L_CCA_EDV) (from CAU images), maintains a commendable AUC of 0.939 (Fig. S11). This streamlined model significantly reduces the number of risk predictors while preserving high prediction accuracy, demonstrating promising potential for practical application in clinical settings. Moreover, the reduced number of risk predictors in the streamlined model alleviates concerns about model overfitting.

## Genetic-imaging integrative analysis – Multi-image risk score (MRS)

Each participant's multi-image risk score (MRS) was calculated as the likelihood of being predicted as a T2D case using XGBoost, which analyzed the medical imaging features for T2D prediction. The odds ratio and its confidence interval for the association between MRS and T2D are shown by percentiles of MRS (Fig. 6D). Compared to the

participants in the 40–60% MRS decile group, the risk of T2D increased with MRS. Of importance, we further identified that, for the men older than 54 years old with a family history of T2D, the case vs. control ratio of sample size was 9.3 in the 90–100% MRS decile group, much higher than 1.3, which MRS was not considered (Fig. 6E).

### Online T2D-risk assessment

We have established a website where users can calculate their T2D risk online. To obtain the risk assessment, users are required to provide age, sex, and family history of T2D, and they can optionally provide PRS and MRS (Fig. 6F). PRS and MRS can be entered manually or uploaded as a file (Supplemental Text 1). Additionally, we have provided PRS and MRS risk percentages based on the study population as a reference. The online risk assessment offers information, including the risk of developing T2D over 3, 5, and 7 years, T2D-free probability, and T2D risk with and without considering PRS (Fig. 6G). The assessment takes into account both PRS and MRS (Fig. 6G). For example, consider a 50-year-old male with a family history of T2D and PRS 1.5 and MRS 1.5. Without considering PRS, the risk (probability) of developing T2D after a 7-year follow-up is 0.23. However, when PRS is considered, the risk increases to 0.37. Furthermore, considering MRS further increases the risk to 0.81. The online tool provides these valuable insights to users based on their input data.

## Discussion

In this study, we conducted a comparative analysis of two prediction models based on GWAS data. The first method utilized T2D-associated SNPs derived from our GWAS with a limited sample size, either as individual predictors or in combination to construct a PRS for the prediction model. The second method incorporated T2D-associated SNPs from previously published GWASs with a significantly larger sample size or utilized summary statistics of whole-genome SNPs from GWASs with a considerably larger sample size to construct the PRS. Notably, the latter approach yielded a higher prediction AUC. These findings underscore the substantial impact of sample size in GWAS, PRS construction, and subsequent classification and prediction analyses, aligning with previous research[39]. Consequently, in situations where the sample size is limited, we propose utilizing external genetic information such as SNPs and summary statistics from published studies with larger sample sizes, which not only facilitates the development of a more predictive PRS and model but also reduces computational overhead[40].

Our study investigated the importance of employing a precision phenotype definition to evaluate disease risk. We also addressed a potential limitation associated with the prevalent use of self-reported disease status. Utilizing four T2D definitions, Type IV, which integrates self-reported T2D with measurements of HbA1c and GLU-AC, emerged as a definition closely aligned with clinical practice. Our results demonstrate that the model based on T2D Definition IV exhibits the highest prediction accuracy. Consequently, in this study, superior diagnostic accuracy corresponds with higher prediction accuracy. Furthermore, the application of self-reported T2D (Definition Type I) yields an AUC significantly lower than the AUC of Definition Type IV. This outcome underscores a potential limitation associated with the commonly used self-report disease status, which functions as a convenient phenotype in the analysis of TWB data.

Our study emphasizes the superiority of disease family history as a predictor of T2D compared to T2D-associated SNPs and PRS. The inclusion of genetic factors such as significant SNPs and PRS as additional predictors, given family history, only results in modest improvements in the model's predictive capability. Family history encompasses genetic, epigenetics, and shared environmental influences, which are crucial in understanding the etiology of T2D[41]. Additionally, we observed that the disease history of siblings provides more informative value for prediction than the disease history of parents[42].

T2D subgrouping can facilitate the implementation of precision medicine in clinical practice, particularly when utilizing complex data[43]. This study demonstrated a positive association between PRS and MRS with T2D risk. Notably, we identified a high-risk subgroup of women older than 59 years with a family history of T2D, where the case vs. control ratio of sample size in the 80–100% PRS decile group ranged from 7 to 13, significantly higher than the overall population. Similarly, for MRS, we found a high-risk subgroup of men older than 54 years with a family history of T2D, where the case vs. control ratio of sample size in the 90–100% MRS decile group was 9.3, considerably higher than the ratio of 1.3 when MRS was not considered. These results demonstrate the utility of PRS and MRS in identifying high-risk subgroups for T2D.

In the PRS-CSx method, we considered three weighting methods to combine several population-specific PRSs into the final PRS: (1) an equal weight, (2) the population-specified weight, and (3) the meta-effect size for each SNP. Our results showed that the meta-effect size obtained a worse performance. The population-specified weight performed best; however, the result may vary between cohorts.

In this study, our PRS based on PRS-CSx achieved an AUC of 0.732 for T2D prediction. The AUC increased to 0.915 after further, including age, sex, and family history of T2D in the prediction model. When comparing our results with the previous publications, Khera, Chaffin[44] achieved an AUC of 0.725 using a logistic regression that included age, sex, and PRS constructed with LDpred[45]. Imamura, Shigemizu[14] achieved an AUC of 0.648 with a PRS built by 49 T2D-associated SNPs with LD weights, and the AUC increased to 0.787 after including age, sex, and BMI. Ge, Irvin[18] achieved an AUC of 0.694 with a PRS constructed using summary statistics from three large-scale GWASs. Walford, Porneala[46] achieved an AUC of 0.641 with a PRS built by 63 SNPs, age, and sex. In summary, our study utilized phenotype refinement through HbA1c and fasting glucose, employed XGBoost with superior performance, and considered the family history of T2D as a critical factor for T2D prediction, leading to improved performance compared to previous studies.

Including environmental factors such as education level, drinking level, exercise habit, and the number of exercise types in our models increased prediction accuracy for non-T2D participants but decreased accuracy for T2D cases. The overall improvement in prediction performance achieved by including these environmental factors was relatively modest and did not reach statistical significance. Similarly, including SNP-SNP interactions in the models did not lead to a significant improvement. While SNP-SNP interactions have been proposed as a potential explanation for missing heritability[47], our findings indicate that incorporating these interactions does not provide additional benefits when PRS is already included in the model. This could be attributed to PRS already capturing a substantial portion of the genetic component, making incorporating SNP main effects and SNP-SNP interactions less impactful.

This study demonstrates good ability in detecting T2D cases, but we observed that some self-reported non-T2D individuals might be misclassified as T2D cases. Further investigation suggested that these cases represent individuals in a pre-T2D stage. Firstly, their T2D risk at the follow-up time was higher than true non-T2D participants but lower than the confirmed T2D cases, indicating an elevated but not fully developed risk. Secondly, these individuals exhibited higher HbA1c and fasting glucose levels than true non-T2D participants, albeit lower than confirmed T2D cases, suggesting a pre-T2D stage. Lastly, when redefining the phenotype using HbA1c and fasting glucose, most of these participants did not meet the inclusion criteria for the control group, further suggesting that they may not be genuinely non-T2D participants. Considering these factors, it is evident that although these participants are self-reported as non-T2D, they are likely in a pre-T2D stage, with an increased risk of developing T2D in the future. It is crucial to follow up with these individuals, monitor their condition

closely, and implement preventive interventions to mitigate the risk of T2D development.

The integration of genetics and medical imaging data into risk assessment shows excellent potential for enabling early T2D detection and prevention, albeit at a higher cost. Practical examples from health examinations and screenings, such as the MJ Health Survey Database in Taiwan[48], provide compelling evidence for successfully incorporating these data into real-world practices. These examples highlight the valuable role that genetics and medical imaging data can play in enhancing risk assessment and underscore the potential benefits of integrating these approaches for improved disease management and prevention. Notably, the performance of T2D classification and prediction in the established models was validated in a second independent dataset, yielding equally impressive results, thus demonstrating that the results are not due to overfitting.

While our current study presents a robust proof-of-principle model for disease risk evaluation based on genetic and multi-modality medical imaging variables within the TWB, we recognize the importance of external validation for broader generalizability. However, the current landscape poses challenges in accessing publicly available datasets encompassing genetic and all four types of medical imaging data. Despite the inherent limitations in readily available datasets with comparable characteristics, we are actively collaborating with a medical center to collect external validation data for future studies.

Another limitation of our study is that due to limited follow-up time in the TWB, only a limited number of participants experienced a change in T2D status from baseline to follow-up, particularly for redefining the phenotype using HbA1c and fasting glucose. To assess the early detection capability of our model for T2D, we are currently addressing this issue by monitoring the participants who exhibited changes in self-reported T2D status from baseline to follow-up in our Cox regression model. This limitation can be overcome in future studies as the TWB continues to track these samples. In addition, conducting a cohort survey or clinical trial is warranted to evaluate the high-risk subgroups our PRS and MRS identified for future precision T2D medicine.

In conclusion, our study surpassed previous research in predicting and classifying T2D. We successfully developed artificial intelligence models that effectively combined genetic markers, medical imaging features, and demographic variables for early detection and risk assessment of T2D. PRS and MRS were instrumental in identifying high-risk subgroups for T2D risk assessment. To facilitate online T2D risk evaluation, we have also created a dedicated website.

## Methods

### Inclusion and ethics declarations

The TWB collected written informed consent from all participants. The TWB (TWBR10911-01 and TWBR11005-04) and the Institute Review Board at Academia Sinica approved our data application and use (AS-IRB01-17049 and AS-IRB01-21009).

### Study participants and variables

This study included a genetic-centric analysis (Analysis 1) and a genetic-imaging integrative analysis (Analysis 2). A total of 68,911 participants in the TWB were analyzed.

In the genetic-centric analysis, 50,984 participants who had only baseline data (i.e., without follow-up data) were used as the training and validation samples; they consisted of 2531 self-reported T2D patients and 48,453 self-reported non-T2D controls (Fig. 1A and S1). We assigned 80% and 20% of data as the training and validation samples. The GWAS samples (Dataset 1, $N = 35,688$), training samples (Dataset 2, $N = 12,236$; Dataset 4, $N = 40,787$), and validation samples (Dataset 3, $N = 3060$; Dataset 5, $N = 10,197$). For classification analysis, testing samples comprised two independent datasets: Dataset 6 ($N = 8827$) and Dataset 7 ($N = 936$), while for prediction analysis, they were

represented as Datasets 6' ($N = 8827$) and Dataset 7' ($N = 936$) (Fig. 1A and Fig. S1). Here, 9763 participants who had both baseline and follow-up data were used as the testing samples, where 8,827 and 936 participants were recruited as the first and second testing datasets; they consisted of 528 self-reported T2D patients and 9235 self-reported non-T2D controls at baseline; 767 self-reported T2D patients and 8996 self-reported non-T2D controls at follow-up (Fig. 1A and Fig. S1A).

In addition to the self-reported T2D, hemoglobin A1C (HbA1c) and fasting glucose (GLU-AC) collected in both baseline and follow-up were used to refine the self-reported T2D phenotype. In total, we considered four definitions for T2D as follows: (1) Self-reported T2D: The case and control information was collected from the questionnaire directly; (2) Self-reported T2D + HbA1C: A case was defined as self-reported T2D and HbA1C ≥ 6.5% and control was defined as self-reported non-T2D and HbA1C ≤ 5.6%; (3) Self-reported T2D + GLU-AC: A case was defined as self-reported T2D and GLU-AC ≥ 126 and control was defined as self-reported non-T2D and GLU-AC < 100; (4) Self-reported T2D + HbA1c + GLU-AC: A case was defined as self-reported T2D, HbA1C ≥ 6.5%, or GLU-AC ≥ 126 and control was defined as self-reported non-T2D, HbA1C ≤ 5.6%, and GLU-AC < 100 (Figs. 1B and 1C).

The demographic characteristics of the study population for four phenotype definitions were shown (Table 1). The table reveals that the participants in the T2D case group are older than those in the control group, with a higher proportion of males.

Other variables in the genetic-centric analysis (Fig. 1D) are illustrated as follows: Demographic variables included age, sex, and family history of T2D. Four types of family history were: T2D occurrence in any of father and mother (parents) (Yes or No), in any of brother and sister (sibs) (Yes or No), in any of father, mother, brother, and sister (Yes or No), and the number of T2D cases in father, mother, brother, and sister (0, 1, 2, 3, or 4). Environmental exposures included education level, drinking level, exercise habits, and the number of exercise types.

Whole-genome genotyping using one of two SNP arrays was performed based on the samples in the baseline. TWBv1.0 SNP array with approximately 650,000 SNP markers or TWBv2.0 SNP array with approximately 750,000 SNP markers was employed. Imputation was performed based on the 1KG-EAS panel[49]. The SNPs with an info score of less than 0.9 were removed[50]. Sample and marker quality controls followed the procedures of Yang, Chu[51]. Related samples were removed using the index of identity by descent in the quality control procedure. External information about T2D-associated SNP sets, and effect sizes based on the GWAS summary statistics of T2D were collected (Supplemental Text 2).

In the genetic-imaging integrative analysis, 17,785 participants who had both genetic data and medical imaging data were analyzed (Fig. 2A and S1B); they consisted of 1366 self-reported T2D patients and 16,419 self-reported non-T2D controls (Fig. 2A); here, the case and control were defined based on the questionnaire at follow-up rather than baseline. For example, based on the T2D Definition IV (Fig. 1C), 7786 participants, which consisted of 1118 cases and 6668 controls, were analyzed (Fig. 2A). The entire dataset was split into training + validation and testing sets at an 8:2 ratio. Subsequently, the training + validation set was further randomized into distinct training and validation datasets, maintaining an 8:2 ratio. Imaging report variables in the genetic-imaging integrative analysis (Upper left in Fig. 2A) consisted of 28 ABD features, 29 CAU features, 85 BMD features, and 10 ECG features (Supplemental Data 1). TU features were not included because of a small sample size. The details about the medical imaging protocol can be referred to TWB (https://www.biobank.org.tw/about_value.php). In the flowchart of PRS calculation, external information about T2D-associated SNP sets and GWAS summary statistics from DIAGRAM are provided (Fig. 2B). This paper provides only numerical data in aggregate and summary statistics. No individuals can be identified.

**Table 1 | Demographic characteristics of the study population for four T2D definitions**

| Phenotype | | Overall | T2D group | Non-T2D group | |
|---|---|---|---|---|---|
| Def I | N | 59,811 | 3011 | 56,800 | |
| | Age, mean (SD) (years) | 50.57 (10.57) | 58.30 (7.62) | 50.16 (10.54) | $p \approx 0$ |
| | Age, median (years) | 52 | 60 | 51 | |
| | Sex, male, n (%) | 18,973 (31.72%) | 1350 (44.83%) | 17,623 (31.02%) | $p = 1.48 \times 10^{-56}$ |
| Def II | N | 30,227 | 2320 | 27,907 | |
| | Age, mean (SD) (years) | 47.87 (10.71) | 58.22 (7.66) | 47.01 (10.48) | $p \approx 0$ |
| | Age, median (years) | 48 | 59 | 46 | |
| | Sex, male, n (%) | 8962 (29.64%) | 1055 (45.47%) | 7907 (28.33%) | $p = 2.11 \times 10^{-67}$ |
| Def III | N | 32,520 | 1155 | 31,365 | |
| | Age, mean (SD) (years) | 49.25 (10.67) | 58.28 (7.55) | 48.91 (10.62) | $p = 1.36 \times 10^{-235}$ |
| | Age, median (years) | 50 | 59 | 50 | |
| | Sex, male, n (%) | 8987 (27.63%) | 537 (46.49%) | 8450 (26.94%) | $p = 5.07 \times 10^{-48}$ |
| Def IV | N | 20,545 | 2393 | 18,152 | |
| | Age, mean (SD) (years) | 47.95 (10.84) | 58.24 (7.63) | 46.59 (10.47) | $p \approx 0$ |
| | Age, median (years) | 48 | 59 | 46 | |
| | Sex, male, n (%) | 5746 (27.96%) | 1,091 (45.59%) | 4,655 (25.64%) | $p = 1.36 \times 10^{-92}$ |

Sample size, mean and median age with standard deviation (SD), the number and proportion of males, and their differences in the T2D case and control groups are provided. *P*-values for two-sample two-sided t-test and chi-square test are displayed. Note that the total sample size in this table is *N* = 59,811, comprising 40,787 participants in the training dataset, 10,197 participants in the validation dataset, and 8827 participants in the first testing dataset (refer to Fig. 1). Note that the participants in the second independent testing dataset (*N* = 936) (refer to Datasets 7 and 7′ in Fig. 1) are only used for replication purposes and are therefore not included in this table.

## Polygenic risk score for T2D

PRS was constructed by using PRS-CS[52] and PRS-CSx[19]. PRS-CS[52] was run based on the meta-GWAS summary statistics of T2D in East Asia in the DIAGRAM Consortium[53] and the linkage disequilibrium (LD) reference from the EAS population of the 1000 Genomes Project[49]. PRS was calculated using PLINK (--score command) based on our genotype data, and 884,327 SNP effects were estimated using PRS-CS. Normalized PRS was standardized to mean = 0 and standard deviation = 1. PRS-CSx[19] based on the meta-GWAS summary statistics of T2D in multiple populations, including (a) East Asian: 56,268 cases and 227,155 controls in the DIAGRAM Consortium[53]; (b) European: 80,154 cases and 853,816 controls in the DIAGRAM Consortium[53]; (c) South Asian: 16,540 cases and 32,952 controls in the DIAGRAM Consortium[53], and the LD reference from each of the three populations (EAS, EUR, and SAS). 884,327, 880,098, and 900,047 SNPs for EAS, EUR, and SAS were applied to our data to calculate the population-specific PRS for each individual using the PLINK (--score command). We combined the three population-specific PRS with equal weight to calculate a final PRS. R language was used to standardize the PRS to mean = 0 and standard deviation = 1.

## Classification and prediction for T2D

XGBoost algorithm[29], implemented through Python code, was applied to classify and predict T2D based on the following features: genetic variables, demographic variables, environmental exposures, and imaging report variables. Both classification and prediction models were trained and validated based on the baseline data (Datasets 2–5 in Fig. 1A). Final classification models were built and tested based on the baseline phenotype data (Dataset 6 in Fig. 1A) and further replicated based on the second independent testing dataset (Dataset 7 in Fig. 1A). Final prediction models were built and tested based on the follow-up phenotype data (Dataset 6′ in Fig. 1A) and replicated based on the second independent testing dataset (Dataset 7′ in Fig. 1A). The illustration of the data used for classification and prediction tasks in Analysis 1 and Analysis 2 are provided (Supplementary Table S1).

The XGBoost models were trained with the following default parameter settings: a maximum tree depth of 6, a learning rate of 0.3, a regularization parameter alpha (L1) of 0, a regularization parameter lambda (L2) of 1, 100 boosting stages, and an early-stop parameter of 30. Feature importance was calculated based on the average of three importance metrics: weight, gain, and cover indices for each variable within a single tree and then averaged across all the trees in a model[54]. Feature selection was performed based on the feature importance score. Parameter tuning was conducted to establish the best model (Supplementary Table S4).

The area under the receiver operating curve (AUC) was calculated to evaluate the model's overall performance. The two-sided DeLong test (DeLong et al., 1988) examined AUCs' differences. Bonferroni's correction[55] was applied to control for a family-wise error rate in multiple comparisons. In the best model, accuracy, sensitivity, specificity, and F1-score were calculated to evaluate the performance, where the optimal cut-off value of the XGBoost model was calculated using the Youden index[56] in the validation data.

## Event history analysis and online risk assessment

In the genetic-centric analysis, multivariate Cox regression[57] was applied to identify important risk factors for the T2D event time and estimate the T2D-free probability in the testing datasets. The event was defined as the occurrence of T2D in the follow-up for non-T2D participants at baseline. The Cox regression analysis considered two types of time scales (i.e., time-on-study and age) and three types of sex variable treatment (i.e., adjusting for sex as a covariate, conducting sex-specific analysis with an assumption of a common sex effect, and performing sex-specific analysis with different sex effects), resulting in six analyses (refer to Supplementary Table S3).

The initial three analyses considered time-on-study as the time scale, with age at baseline included as a covariate, and incorporating the following sex variable treatment: (1) Model 1: Sex was treated as a covariate in the analysis; (2) Model 2: Sex-specific analysis, assuming a common effect for males and females; (3) Model 3: Sex-specific analysis with different effects for males and females. The subsequent three analyses considered age as the time scale, with age-at-baseline as left truncation, along with the three sex variable treatments, similar to the time-on-study analysis, to be Models 4 – 6.

The time-on-study scale analysis calculated the event time as the duration from baseline to follow-up. In the age scale analysis, the event

time was age-at-follow-up. In each model, the median event time in weeks was also calculated. Because medical imaging data were only available in the follow-up, the genetic-imaging integrative analysis applied multivariate logistic regression[58] to identify important risk factors for T2D events and estimate the T2D-free probability in the testing datasets. In addition, we established a website at https://hcyang.stat.sinica.edu.tw/software/T2D_web/header.php to provide an online risk assessment for T2D.

## Web resources

We established a website at https://hcyang.stat.sinica.edu.tw/software/T2D_web/header.php to provide an online risk assessment for T2D.

## Reporting summary

Further information on research design is available in the Nature Portfolio Reporting Summary linked to this article.

## Data availability

The data analyzed in this study were obtained from the Taiwan Biobank with proper approval. As the data are subject to ownership rights held by the Taiwan Biobank, they have not been deposited in a public repository. Researchers interested in accessing the data must do so through a formal application process, subject to approval by the Taiwan Biobank. Detailed instructions on requesting data access can be found on the Taiwan Biobank's official website (https://www.twbiobank.org.tw/index.php). Source data are provided in the Supplementary Information and Source Data files with this paper. In addition to the TWB data, a set of 137 highly significant T2D-associated SNPs from the AGEN can be downloaded from https://blog.nus.edu.sg/agen/summary-statistics/t2d-2020/. Meta-GWAS summary statistics of T2D in multiple populations from the DIAGRAM Consortium can be obtained from https://diagram-consortium.org/downloads.html. The linkage disequilibrium reference from various populations of the 1000 Genomes Project is available for download at https://github.com/getian107/PRScsx. Source data are provided in this paper.

## Code availability

The repository at https://github.com/yjhuang1119/Risk-assessment-model contains code for constructing a disease risk assessment model using eXtreme Gradient Boosting (XGBoost). The code also computes performance metrics for model evaluation and feature importance scores for model explainability. A README is provided.

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

## Acknowledgements

This work was supported by research grants from Academia Sinica (AS-PH-109-01 and AS-SH-112-01). We gratefully acknowledge the Taiwan Biobank for providing the data used in this research and extend our thanks to all its participants for their invaluable contributions. The National Center for Genome Medicine of Taiwan also provided technical support in genotyping. Part of this paper was completed during the first author's master's studies, which were supported by the Mr. Samuel Yin New Students Scholarship and a scholarship offered by the Institute of Statistical Science, Academia Sinica. We thank Mr. Chia-Wei Chen and Dr. Shih-Kai Chu in our research team for providing the genetic data with quality control.

## Author contributions

H.C.Y. conceptualized and supervised the study. Y.J.H. performed data curation and applied software. Y.J.H. & H.C.Y. conducted formal data analysis and result visualization and wrote the paper. C.h.C. & H.C.Y. provided funding acquisition and resources. H.C.Y., Y.J.H., and C.h.C. validated the results.

## Competing interests

The authors declare no competing interests.
