## [Peer Review File · Nature Communications]

AI-Enhanced Integration of Genetic and Medical Imaging Data for Risk Assessment of Type 2 DiabetesREVIEWER COMMENTS

Reviewer #1 (Remarks to the Author):

This manuscript developed a predictive model for the T2D risk prediction. They analyzed data consisting of 68,911 subjects from a Taiwan Biobank. Based on the analysis results, authors argued that the approach integrating genetic and medical imaging information significantly enhanced the T2D risk prediction compared to the model with genetic information. However, its applicability in real world is questionable with its complexity including more than 100 medical image variables, which is not usually available to general population.

1. The definition of T2D case and control needs some discussion. For example, Figure 2 defines control as self-reported no T2D + HbA1c \leq 5.6% or GLU-AC < 100. What if someone is self-reported no T2D + GLU-AC < 100 but also having HbA1c > 6.5%? Based on the current definition, this individual would be classified as control; however, this may be more reasonable to be classified as case.
2. Several information, such as training, testing datasets, T2D definition, PRS construction, etc., is only shown in the figures. Considering their importance to this work, they should be clearly defined or discussed in the text as well.
3. The genetic-centric analysis and the genetic-imaging integrative analysis used different statistical modeling approach. Will it be a fair comparison with different model building approaches by simply using AUC which is more tailored to logistic regression or classification modeling?
4. Please explain why to use several definitions on T2D for the analysis? Please provide their clinical implications corresponding to these different T2D definitions.
5. The T2D definition in line 211 is different from the one defined in Figure 2.
6. In the T2D early detection analysis, authors only select women sample. Please provide the reasoning on such a biased selection. Also, comment on how the generalizability for the performance on male sample. Similar question applies to age threshold for this sample, i.e. why selecting 59 years old. It seems these selections come from nowhere.
7. Line 329. 124 out of 152 imaging features were used in the final model. Will overfitting be

a concern? In addition, is it practical to use so many imaging features for general population?

8. The testing datasets are different for genetic-centric analysis and genetic-image integrative analysis. What if using the same sample?
9. Are there any related sample within TWB? if so, does analysis take sample relatedness into consideration? If not, how will this affect the conclusion?
10. The replication was done still within the same biobank which is not ideal. It would be more convincing to conduct the replication with sample beyond the original biobank cohort.
11. How the hyperparameter tuning was tuned should be provided for reproducibility.
12. The final proposed prediction model outperformed other models with simple demographic variables and PRS. This reviewer is curious how well it could be helpful or applied to T2D early detection in the real world. The higher complexity of the model implies more cost in their application. Is there any trade-off here? Will the cost for this model potentially lower than use traditional glycemic related measurements, such as HbA1c, fasting glucose, and so on?

Minor

13. Line 50 indicated that the prevalence of T2D in Taiwan is around 10%. This is inconsistent to the data shown in line 114 (2531 T2D vs 48450 nonT2D) which is more on the target of 5%, similar to global prevalence. Any explanation on this? Is the argument used in introduction validated? Or is Taiwan biobank a good representative?
14. Need to clarify how sample for training or validation was determined.
15. Lines 358-366. These sentences simply repeated the same thing twice!
16. The study participants and variables section may be more appropriate to be placed within method section.

Reviewer #2 (Remarks to the Author):

Risk assessment and early detection of T2D are vital in improving individuals' health and in this study, authors established a prediction model and identified high-risk subgroups for T2D with 68,911 Taiwan Biobank (TWB) participants. Authors demonstrate that genetic information alone is insufficient for accurate T2D prediction (AUC = 0.73), and incorporating

medical imaging data significantly improves prediction accuracy (AUC = 0.89). The best-performing model integrates genetic, medical imaging, and demographic variables (AUC = 0.94). The study also presents an online risk assessment website for T2D. Manuscript comprehensively evaluated their prediction model. Here are my comments.

1. In this manuscript, authors used GWAS based on 50,984 subjects. There have been multiple GWAS studies using more than 100,000 east asian (for instance, Spracklen et al, Nature, 2000). It seems better to use those larger GWAS summary statistics when authors built PRS and any comparison is required.

2. Authors showed that prediction model including age, sex, family history of T2D, and PRS achieved AUC=0.915. This result is very interesting. It would be interesting to evaluate the importance of each variable. For instance, AUCs from the model with age and without age can be compared and their improvement can be utilized as the variable importance.

3. According to the AUC, the inclusion of age already generates the very large AUC (0.804). It seems that the T2D patients are generally very old, and because of this issue, the prediction accuracy can become very large. If it is true, then I think that prediction accuracy must be evaluated without age.

4. External validation is necessary. For instance, authors can apply to subjects in different cities or the population in the other east asian country.

5. Authors considered four different definitions of T2D. In the final results, authors found that T2D definition IV has the highest accuracy, and it was applied to the Figure 5 and 6. I think that it is better to consider the model with better diagnostic accuracy because the prediction accuracy depends on the characteristics of the selected cohorts. It seems that the data that authors used may include more older subjects.

6. The prediction model and its accuracy must be affected by the sampling error. Authors should provide how to handle this issue.

Minor comments:

1. For https://hcyang.stat.sinica.edu.tw/software/T2D_web/header.php, authors provide the percentage of PRS as a guideline. For instance, 0-10% is -3.488 - -1.28. However I think that those information is not practical because those depends on the number of missing SNP, and the scale, etc.

2. I think that authors should provide the SNPs that they used for calculating PRS and the detailed information about how to calculate PRS.

Reviewer #3 (Remarks to the Author):

This is a thoughtfully done analysis of genetic, clinical, and demographic risk factors for type 2 diabetes (T2D). It is the first such analysis carried out using a unique and valuable data resource from Taiwan.

The work is heavily predictive in focus, and does not question or propose mechanisms relating to the disease. Nevertheless, such predictive analyses are valuable in helping to characterize the genetic contribution to T2D risk. The implications of the short follow-up imaging study are less clear. I appreciate that the authors acknowledge the potential for misclassification of T2D status and conduct some investigations indicating that high-risk people classified as controls may in some sense be pre-diabetic.

Line 31 in the abstract states that the prediction involving genetics, imaging, and demographics identifies subjects at high risk of developing T2D (AUC is a remarkable 0.94). However given the timing of the imaging analysis and T2D diagnosis, it is unclear if this is actually predicting future risk or current T2D status.

Specific comments:

The cohort is insufficiently characterized, e.g. I could not even find the median age and sex

distribution. Is the cohort in any sense representative of the Taiwan population? Are the cases and controls in any sense comparable with respect to age? I assume that the cohort is overwhelmingly Han Chinese but this information should be provided.

The paper lacks context in relation to the known epidemiologic characteristics of type 2 diabetes, even with respect to basic factors such as age and sex differences. Surprisingly there is no mention of BMI in the paper.

I'm not sure how to interpret figure 5b. T2D status at the follow-up time point is an outcome, so it's not clear what is the meaning of stratifying the estimated survival functions on a future event.

Due to the strong age-dependent nature of T2D onset, an alternative to including baseline age as a covariate in the Cox model is to use age as the time variable with age at entry as a left truncation time. Then the baseline hazard would capture age-specific risk. Stratifying by sex, rather than including sex as a covariate is also something to consider. The HR for sex (50% lower risk for females) is remarkably strong and warrants discussion.

The analysis of exercise is limited. At most 290 people report any specific type of exercise. The denominator for these numbers is unclear but it seems to me that exercise may be under-reported in this cohort. The lack of clear results may be due to low power. Testing for multiple types of exercise seems not to have been subject to rigorous multiplicity adjustment.

It would be helpful to consider how the PRS and familial T2D status are related. This may shed light on the contributions of genetics, epigenetics, and shared environment.

I don't really see the point of the on-line tool, it requires people to know their PRS and MRS. How would an individual have access to this information?

Line 275 is confusing, it seems to suggest that HbA1c improves in T2D patients?

In line 425 the claim is too definitive, this could be resolved, e.g. by changing “revealed” to “suggested”.

There seems to be a missing word in line 339 (“which MRS was not considered”)

Authors' Reply

To Reviewer #1 (Remarks to the Author):

This manuscript developed a predictive model for the T2D risk prediction. They analyzed data consisting of 68,911 subjects from a Taiwan Biobank. Based on the analysis results, authors argued that the approach integrating genetic and medical imaging information significantly enhanced the T2D risk prediction compared to the model with genetic information. However, its applicability in real world is questionable with its complexity including more than 100 medical image variables, which is not usually available to general population.

Response: We appreciate the insightful comments provided by the reviewer, and we have addressed the concern raised regarding the clinical application of our developed model. We acknowledge the initial model's limitation in requiring over 100 feature variables, posing challenges for actual clinical implementation, although it has a promising Area Under the Receiver Operating Curve (AUC) of 0.949 in risk prediction. In response, our revised model, featuring only the top eight essential variables in the best model – family history (from the questionnaire), age (from the questionnaire), fatty liver (from ABD images), spine thickness (from BMD images), polygenic risk score (PRS) (from genetic data), end-diastolic velocity in the right common carotid artery (R_CCA_EDV) (from CAU images), RR interval (from ECG images), and end-diastolic velocity in the left common carotid artery (L_CCA_EDV) (from CAU images) – maintains a commendable AUC of 0.939 (refer to **Fig. A1** in this reply letter and **Fig. S11** in the revised manuscript). This streamlined model significantly reduces the number of risk predictors while preserving high prediction accuracy, demonstrating promising potential for practical application in clinical settings.

We add the description in Lines 427 – 442 in the Results section in the revised manuscript as follows: “To address the challenges of practical clinical implementation

in the best XGBoost model, we have proposed an alternative model that requires a limited number of features. We systematically calculated the incremental area under the AUC for each feature by sequentially including those with the highest feature importance, and selected the top features showing a positive AUC increment. The analysis revealed that a sub-model incorporating only the following eight crucial variables: family history (from the questionnaire), age (from the questionnaire), fatty liver (from ABD images), spine thickness (from BMD images), PRS (from genetic data), end-diastolic velocity in the right common carotid artery (R_CCA_EDV) (from CAU images), RR interval (from ECG images), and end-diastolic velocity in the left common carotid artery (L_CCA_EDV) (from CAU images), maintains a commendable AUC of 0.939 (**Fig. S11**). This streamlined model significantly reduces the number of risk predictors while preserving high prediction accuracy, demonstrating promising potential for practical application in clinical settings. Moreover, the reduced number of risk predictors in the streamlined model alleviates concerns about model overfitting.”

Figure A1. Cumulative and incremental AUCs of the top feature variables in the best model. (A) Cumulative AUC. The best prediction model, which contains age, family history of T2D, PRS, and 125 medical imaging feature variables, achieves an AUC of

0.949. In this figure, we list the top 20 feature variables in the best model. The reduced model, comprising only the top eight essential variables – family history (from the questionnaire), age (from the questionnaire), fatty liver (from ABD images), spine thickness (from BMD images), polygenic risk score (PRS) (from genetic data), end-diastolic velocity in the right common carotid artery (R_CCA_EDV) (from CAU images), RR interval (from ECG images), and end-diastolic velocity in the left common carotid artery (L_CCA_EDV) (from CAU images) – maintains a commendable AUC of 0.939. **(B) Incremental AUC (%) for each of the top feature variables.** Feature variables are sorted based on the importance of the feature in the XGBoost model (see **Fig. 6C**). Family history is the most crucial feature, with an AUC of 0.738. Conditional on family history in the model, the inclusion of age results in an increase of the total AUC to 0.834, with an increment of AUC of 0.096. Subsequent sequential inclusion of fatty liver (from ABD images), spine thickness (from BMD images), polygenic risk score (PRS) (from genetic data), end-diastolic velocity in the right common carotid artery (R_CCA_EDV) (from CAU images), RR interval (from ECG images), and end-diastolic velocity in the left common carotid artery (L_CCA_EDV) (from CAU images) yields total AUC values of 0.892, 0.909, 0.925, 0.930, 0.937, and 0.939 with increment values of 0.058, 0.016, 0.017, 0.005, 0.006, and 0.002, respectively.

1. The definition of T2D case and control needs some discussion. For example, Figure 2 defines control as self-reported no T2D + HbA1c \leq 5.6% or GLU-AC $<$ 100. What if someone is self-reported no T2D + GLU-AC $<$ 100 but also having HbA1c $>$ 6.5%? Based on the current definition, this individual would be classified as control; however, this may be more reasonable to be classified as case.

Response:

We appreciate this reviewer's attention. There was a typographical error in **Fig. 2** in the original manuscript, and we acknowledge and apologize for any confusion it may have caused. The correct definition is as follows: Controls are individuals who self-report no T2D, have HbA1c \leq 5.6%, and GLU-AC $<$ 100 mg/dL. Cases are individuals who self-report T2D, have HbA1c \geq 6.5%, or GLU-AC \geq 126 mg/dL. We have now rectified the phenotype definition in **Fig. 2** in the revised manuscript to reflect these criteria accurately.

2. Several information, such as training, testing datasets, T2D definition, PRS construction, etc., is only shown in the figures. Considering their importance to this work, they should be clearly defined or discussed in the text as well.

Response:

We have added the description regarding training, validation, and testing datasets in

Lines 111 – 117 in the Methods section in the revised manuscript as follows: “... We assigned 80% and 20% of data as the training and validation samples, respectively. The GWAS samples (Dataset 1, N = 35,688), training samples (Dataset 2, N = 12,236; Dataset 4, N = 40,787), and validation samples (Dataset 3, N = 3,060; Dataset 5, N = 10,197). For classification analysis, testing samples comprised two independent datasets: Dataset 6 (N = 8,827) and Dataset 7 (N = 936), while for prediction analysis, they were represented as Datasets 6' (N = 8,827) and Dataset 7' (N = 936) (**Fig. 1A** and **Fig. S1**).”

We have added the description regarding the T2D definition in Lines 125 – 133 in the Methods section in the revised manuscript as follows: “In total, we considered four definitions for T2D as follows: (1) Self-reported T2D: The case and control information was collected from the questionnaire directly; (2) Self-reported T2D + HbA1C: A case was defined as self-reported T2D and HbA1C $\geq 6.5\%$ and a control was defined as self-reported non-T2D and HbA1C $\leq 5.6\%$; (3) Self-reported T2D + GLU-AC: A case was defined as self-reported T2D and GLU-AC ≥ 126 and a control was defined as self-reported non-T2D and GLU-AC < 100 ; (4) Self-reported T2D + HbA1c + GLU-AC: A case was defined as self-reported T2D, HbA1C $\geq 6.5\%$, or GLU-AC ≥ 126 and a control was defined as self-reported non-T2D, HbA1C $\leq 5.6\%$, and GLU-AC < 100 (**Figs. 1B** and **1C**).”

We have added the description regarding PRS construction in Lines 171 – 187 in the Methods section in the revised manuscript as follows: “PRS was constructed by using PRS-CS (Ge et al., 2019) and PRS-CSx (Ruan et al., 2022). PRS-CS (Ge et al., 2019) based on the meta-GWAS summary statistics of T2D in East Asia in the DIAGRAM Consortium (Mahajan et al., 2022) and the linkage disequilibrium (LD) reference from the EAS population of the 1000 Genomes Project (Auton et al., 2015). PRS was calculated using PLINK (--score command) based on our genotype data, and 884,327 SNP effects were estimated using PRS-CS. Normalized PRS was standardized to mean = 0 and standard deviation = 1. PRS-CSx (Ruan et al., 2022) based on the meta-GWAS summary statistics of T2D in multiple populations, including (a) East Asian: 56,268 cases and 227,155 controls in the DIAGRAM Consortium (Mahajan et al., 2022); (b) European: 80,154 cases and 853,816 controls in the DIAGRAM Consortium (Mahajan et al., 2022); (c) South Asian: 16,540 cases and 32,952 controls in the DIAGRAM Consortium (Mahajan et al., 2022), and the LD reference from each of the three populations (EAS, EUR, and SAS). 884,327, 880,098, and 900,047 SNPs for EAS, EUR, and SAS were applied to our data to calculate the population-specific PRS for each individual using the PLINK (--score command). We combined the three population-specific PRS with equal weight to calculate a final PRS. R language was used to standardize the PRS to mean = 0 and standard deviation = 1.”

3. The genetic-centric analysis and the genetic-imaging integrative analysis used different statistical modeling approach. Will it be a fair comparison with different model building approaches by simply using AUC which is more tailored to logistic regression or classification modeling?

Response:

In response to the reviewer's comment, we would like to clarify that both the genetic-centric analysis and the genetic-imaging integrative analysis employed the same modeling approach, namely eXtreme Gradient Boosting (XGBoost).

Regarding the choice of performance metric, we acknowledge the reviewer's concern about using AUC, which is commonly associated with logistic regression or classification modeling. In our evaluation of the final model, we recognize the importance of considering multiple performance measures to comprehensively assess its effectiveness. In addition to AUC, we incorporated other metrics, including accuracy, sensitivity, specificity, and F1-score. The result provides a more thorough understanding of the model's capacity, robustness, and potential limitations.

4. Please explain why to use several definitions on T2D for the analysis? Please provide their clinical implications corresponding to these different T2D definitions.

Response:

In response to the reviewer's comment, the utilization of four distinct definitions for Type 2 Diabetes (T2D) in our analysis serves two primary purposes. Firstly, it allows us to explore the significance of employing a precision phenotype definition in the evaluation of disease risk. Additionally, we aim to highlight a potential limitation associated with the prevalent use of self-reported disease status as a convenient phenotype in the analysis of the Taiwan Biobank data.

Among the four T2D definitions utilized, Type IV, which combines self-reported T2D with measurements of HbA1c and GLU-AC, offers a definition that closely aligns with clinical practice. Our results show that the model based on the T2D Definition IV has the highest prediction accuracy (refer to **Fig. 3B** in the original and revised manuscripts). Therefore, in this study, better diagnostic accuracy and higher prediction accuracy are actually consistent.

In addition, the application of self-report T2D (Definition Type I) has an AUC significantly lower than the AUC of Definition Type IV (refer to **Fig. 3B** in the original and revised manuscripts). The result highlights a potential limitation associated with the commonly used self-report disease status, which serves as a convenient phenotype in analyzing Taiwan Biobank data.

We appreciate the reviewer's insightful comment, and we believe that the inclusion of various T2D definitions enhances the robustness of our analysis by shedding light on the importance of precision phenotyping in disease risk assessment. We have added a new paragraph in Lines 485 – 496 in the Discussion section in the revised manuscript as follows: "In our study, we investigated the importance of employing a precision phenotype definition for the evaluation of disease risk. We also addressed a potential limitation associated with the prevalent use of self-reported disease status. Utilizing four T2D definitions, Type IV, which integrates self-reported T2D with measurements of HbA1c and GLU-AC, emerged as a definition closely aligned with clinical practice. Our results demonstrate that the model based on T2D Definition IV exhibits the highest prediction accuracy, despite the presence of some missing data for HbA1c or GLU-AC. Consequently, in this study, superior diagnostic accuracy corresponds with higher prediction accuracy. Furthermore, the application of self-reported T2D (Definition Type I) yields an AUC significantly lower than the AUC of Definition Type IV. This outcome underscores a potential limitation associated with the commonly used self-report disease status, which functions as a convenient phenotype in the analysis of Taiwan Biobank data."

5. The T2D definition in line 211 is different from the one defined in Figure 2.

Response:

We appreciate this reviewer's attention. There was indeed a typographical error in **Fig. 2** in the original manuscript, and we acknowledge and apologize for any confusion it may have caused. The correct definition is as follows: Controls are individuals who self-report no T2D, have HbA1c $\leq 5.6\%$, and GLU-AC < 100 . Cases are defined as individuals who self-report T2D, have HbA1c $\geq 6.5\%$, or GLU-AC ≥ 126 . We have now rectified the phenotype definition in **Fig. 2** in the revised manuscript to reflect these criteria accurately.

6. In the T2D early detection analysis, authors only select women sample. Please provide the reasoning on such a biased selection. Also, comment on how the generalizability for the performance on male sample. Similar question applies to age threshold for this sample, i.e. why selecting 59 years old. It seems these selections come from nowhere.

Response:

In response to the reviewer's comment regarding the selective inclusion of women in the T2D early detection analysis, we would like to clarify that our study involved performing a stratified analysis across various combinations of age subgroups, sex

subgroups, and family history subgroups to identify high-risk subgroups (refer to **Fig. A2** in this reply letter and **Fig. S7** in the revised manuscript).

Age was stratified into four subgroups based on quartiles: 0–25%, 25%–50%, 50%–75%, and 75%–100%, corresponding to age subgroups of ≤ 43 , 43–52, 52–59, and >59 years of age, respectively. Among the 16 analyses conducted (comprising the combination of 4 age subgroups, 2 sex subgroups, and 2 family history subgroups), only the female group aged >59 with a family history demonstrated a robust and reliable finding, indicating a significantly higher risk compared to the reference group. We have added the description regarding stratified analysis in Lines 333 – 343 in the Results section in the revised manuscript as follows: “In addition, we performed a stratified analysis across various combinations of age subgroups, sex subgroups, and family history subgroups to identify high-risk subgroups, where age was stratified into four subgroups based on quartiles: 0–25%, 25%–50%, 50%–75%, and 75%–100%, corresponding to age subgroups of ≤ 43 , 43–52, 52–59, and >59 years of age, respectively (**Fig. S7**) ... Due to ambiguity or instability in the evidence for other combinations, we chose not to report them.”

Regarding the generalizability of our findings to male samples, the evidence is not as conclusive. Although we observed a high risk of 10.0–12.0-fold in the top 20% decile Polygenic Risk Score (PRS) group for males aged >59 with a family history (refer to **Fig. A3** in this response letter), it is noteworthy that the male reference group, not accounting for PRS risk, also exhibited a moderately high disease risk of 7.3-fold. Due to the ambiguity in the evidence, we chose to report the most reliable high-risk group identified, specifically the high-risk subgroup of women aged over 59 with a family history of T2D, in the main manuscript.

Similarly, except for >59 years of age, other age subgroups did not show stable evidence for other combinations, and we chose not to report them.

Figure A2. Identification of high-risk groups. A stratified analysis across various combinations of age subgroups, sex subgroups, and family history subgroups was performed to identify high-risk subgroups. In each chart, the figures from the inner to the outer represent (i) the case-to-control ratio, (ii) the number of cases, and (iii) the number of controls in the PRS decile subgroups.

Figure A3. Sex-specific analysis for identification of high-risk subgroups. (A) Male; (B) Female. In the chart, the figures from the inner to the outer represent (i) the case-to-control ratio, (ii) the number of cases, and (iii) the number of controls in the PRS decile subgroups. A high-risk group was identified, comprising females older than 59 with a T2D family history and falling into the PRS group in the 80%–100% decile group. We identified a female group aged > 59 with family history showed a reliable finding that a very high risk compared to the reference group. Although a male group aged >59 with family history showed a high risk of 10–12 but the reference group also has a high risk of 7.3. The evidence is not particularly strong. Therefore, we only reported the most reliable high-risk subgroup.

7. Line 329. 125 out of 152 imaging features were used in the final model. Will overfitting be a concern? In addition, is it practical to use so many imaging features for general population?

Response:

In response to the reviewer’s comment on Line 329 in the original manuscript regarding the utilization of 125 out of 152 imaging features in the final model, we appreciate the concern raised about the potential for overfitting and the practicality of employing such a large number of imaging features for the general population.

Addressing the overfitting concern, the robustness of our model is supported by its consistent high performance on two independent testing datasets, as evidenced by high Area Under the Receiver Operating Curve (AUC) and accuracy. These results indicate that overfitting is not a significant issue in our study. While external validation would be helpful, unfortunately, the data is currently not available.

Regarding more than 100 imaging features in the final model, we concur with this reviewer that this will pose challenges for actual clinical implementation, although it has a promising AUC of 0.949 in risk prediction. In response, our revised model, featuring only the top eight essential variables – family history (from the questionnaire), age (from the questionnaire), fatty liver (from ABD images), spine

thickness (from BMD images), polygenic risk score (PRS) (from genetic data), end-diastolic velocity in the right common carotid artery (R_CCA_EDV) (from CAU images), RR interval (from ECG images), and end-diastolic velocity in the left common carotid artery (L_CCA_EDV) (from CAU images) – maintains a commendable AUC of 0.939 (refer to **Fig. A1** in this reply letter and **Fig. S11** in the revised manuscript). This streamlined model significantly reduces the number of risk predictors while preserving high prediction accuracy, demonstrating promising potential for practical application in clinical settings. Moreover, the reduced number of risk predictors in the streamlined model alleviates concerns about model overfitting.

8. The testing datasets are different for genetic-centric analysis and genetic-image integrative analysis. What if using the same sample?

Response:

In response to the reviewer's query about the use of different training, validation, and testing datasets for the genetic-centric analysis and genetic-image integrative analysis, we conducted additional analyses using the same sample to assess the comparability of results.

For instance, when applying the same training, validation, and testing data used in the genetic-image integrative analysis in the original manuscript to both the genetic-image integrative analysis and the genetic-centric analysis, we observed reasonable consistency in the results. In the genetic-image integrative analysis, the model incorporating genetic variables achieved an Area Under the Curve (AUC) of 0.68, while the model including both genetic and demographic variables achieved an improved AUC of 0.87. Similarly, in the genetic-centric analysis using the same dataset, the model with genetic variables achieved an AUC of 0.73, and the model integrating both genetic and demographic variables attained an AUC of 0.89.

These comparable results support the robustness of our findings. Notably, the higher AUC observed in the analysis incorporating both genetic and demographic variables is attributed to the larger sample size available for this particular analysis. In the genetic-centric analysis, the training, validation, and testing datasets had sample sizes of 40,787, 10,197, and 8,827, respectively (see **Fig. 1** in the revised manuscript). Conversely, the genetic-image integrative analysis utilized smaller sample sizes of 4,689, 1,175, and 1,469 for the training, validation, and testing datasets, respectively (see **Fig. 2** in the revised manuscript). This variation is due to the availability of medical image data only during follow-up, leading to distinct datasets for the genetic-centric and genetic-image integrative analyses. We opted for this approach to maximize sample utilization while considering the constraints posed by the availability of medical image data.

9. Are there any related sample within TWB? if so, does analysis take sample relatedness into consideration? If not, how will this affect the conclusion?

Response:

Yes, there are related samples within TWB. However, our quality control procedures for genetic data have already addressed this issue. As mentioned in the section “Study participants and variables” in the manuscript, sample and marker quality controls followed the procedures of Yang et al. (2016). We addressed relatedness by identifying samples with identity by descent (IBD) (PLINK --genome) and retained only one sample from each related pair having an estimated IBD of >0.1875 . The selection criteria prioritized samples with a higher genotyping call rate. Consequently, our subsequent analysis is free from related samples, ensuring that the sample relatedness will not affect the conclusion. We have added the description in Lines 150 – 151 in the Methods section in the revised manuscript as follows: “Related samples were removed using the index of identity by descent in the quality control procedure.”

10. The replication was done still within the same biobank which is not ideal. It would be more convincing to conduct the replication with sample beyond the original biobank cohort.

Response:

We appreciate the reviewer’s insightful suggestion for external validation, and we acknowledge the importance of further validation with an external dataset. Integrative studies involving genetic and multi-modality medical imaging variables for disease risk evaluation are at the forefront of research and necessitate time to accumulate sufficient data.

While we agree that external validation with a diverse dataset would be ideal, it is essential to recognize the current limitations in the availability of publicly accessible datasets that simultaneously include genetic data and all four types of medical imaging data, especially for Asian populations. Our study, uniquely positioned within the Taiwan Biobank, fills a critical gap by providing a comprehensive analysis for the Asian population, contributing significantly to the scarcity of Asian data in global research. The distinctive nature of our dataset, not readily available in other biobanks or initiatives like the UK Biobank, underscores the pioneering nature of our study.

We believe that our proof-of-principle study, though conducted within the Taiwan Biobank, is forward-looking and emphasizes the promising trend of combining genetic and medical imaging data for future disease risk evaluation. While we acknowledge the evolving landscape and anticipate the gradual emergence of datasets with similar characteristics, waiting for their availability could substantially

delay critical discoveries. In our discussion, we mention the potential for datasets of this nature to become available gradually, with self-funded health checkups being a potential source for such tests.

To enhance the robustness of our findings, we have rigorously replicated our model using two independent testing datasets within the Taiwan Biobank, as extensively detailed in the paper. Additionally, we are actively pursuing collaboration with a medical center, seeking Institutional Review Board (IRB) approval to collect external validation data. However, this process is time-consuming, and we anticipate delays in obtaining and incorporating external datasets for future validation.

We have incorporated a new paragraph into the Discussion section of the revised manuscript in Lines 572 – 579 to highlight the significance of external validation for our findings as follows: “While our current study presents a robust proof-of-principle model for disease risk evaluation based on genetic and multi-modality medical imaging variables within the Taiwan Biobank, we recognize the importance of external validation for broader generalizability. However, the current landscape poses challenges in accessing publicly available datasets that encompass both genetic and all four types of medical imaging data. Despite the inherent limitations in readily available datasets with comparable characteristics, we are actively engaged in collaborations with a medical center to collect external validation data for future studies.”

We sincerely appreciate the reviewer’s understanding of the complexities involved in accessing external datasets and our commitment to advancing precision medicine. We are dedicated to further validating our model and will actively work towards incorporating external datasets in subsequent studies. Your feedback is invaluable in strengthening the integrity and applicability of our work, and we are eager to address any additional concerns or suggestions you may have.

11. How the hyperparameter tuning was tuned should be provided for reproducibility.

Response:

We illustrated the hyperparameter tuning and comparison between prediction models using default parameters and tuned parameters in **Table S3** in the original and revised manuscripts. A grid search method was applied to optimize several parameters in the best model. The tuned parameters encompassed the learning rate (0.001, 0.01, 0.3), the minimum loss reduction required for splitting ($\gamma = 0, 0.1, 1, 1.5$), the maximum depth of a tree ($\text{max_depth} = 3, 6, 9$), the minimum sum of instance weight required in a child ($\text{min_child_weight} = 1, 5, 10$), the subsample ratio of training instances ($\text{subsample} = 0.8, 0.9, 1$), and the control of the balance between positive and negative weights ($\text{scale_pos_weight} = 0, 1, 2, 4$). The default parameters were set

as follows: learning rate = 0.3, gamma = 0, max_depth = 6, min_child_weight = 1, subsample = 1, and scale_pos_weight = 1. Out of the 1,296 training combinations, the optimal parameters were determined to be: learning rate = 0.3, gamma = 1.5, max_depth = 3, min_child_weight = 1, subsample = 0.8, and scale_pos_weight = 2. Because of a very limited difference in the results between prediction models using default and tuned parameters. The default parameters and results provided for reproducing in the manuscript.

12. The final proposed prediction model outperformed other models with simple demographic variables and PRS. This reviewer is curious how well it could be helpful or applied to T2D early detection in the real world. The higher complexity of the model implies more cost in their application. Is there any trade-off here? Will the cost for this model potentially lower than use traditional glycemetic related measurements, such as HbA1c, fasting glucose, and so on?

Response:

Thank you for the thoughtful comment. Of importance, our study focuses on precision medicine, aiming to identify high-risk subgroups rather than adopting a one-size-fits-all approach. Compared to the traditional glycemetic measurements, leveraging genetic and medical imaging data provides more detailed insights into individual characteristics, enabling personalized risk assessment.

In addition, traditional glycemetic measurements like HbA1c and fasting glucose have demonstrated limitations in disease risk prediction, as highlighted in various studies (Buchanan et al., 2009; Eehalt et al., 2017; Greenhalgh et al., 2011; John, 2022; Li et al., 2018; Nowicka et al., 2011; Panwar et al., 2013; Spiller et al., 2018). These metrics often show limited sensitivity, especially in specific populations, and lack accuracy in predicting pre-diabetes and T2D.

Acknowledging the trade-off between prediction accuracy and cost, our analysis indicates a viable compromise. A refined model, featuring only eight essential variables – family history, age, fatty liver, spine thickness, polygenic risk score (PRS), end-diastolic velocity in the right common carotid artery (R_CCA_EDV), RR interval, and end-diastolic velocity in the left common carotid artery (L_CCA_EDV) – maintains a commendable AUC of 0.939 (refer to **Fig. A1** in this reply letter and **Fig. S11** in the revised manuscript). This streamlined model significantly reduces the number of predictors and associated data collection costs while preserving high prediction accuracy. This finding suggests promising potential for practical application in precision medicine of T2D.

Minor

13. Line 50 indicated that the prevalence of T2D in Taiwan is around 10%. This is inconsistent to the data shown in line 114 (2531 T2D vs 48450 nonT2D) which is more on the target of 5%, similar to global prevalence. Any explanation on this? Is the argument used in introduction validated? Or is Taiwan biobank a good representative?

Response:

The data presented in Line 114 (2,531 T2D vs. 48,450 non-T2D) was based on T2D Definition I. As noted earlier, Definition IV provides a more precision definition for T2D. When applying Definition IV, the numbers of T2D cases and controls are 2,054 and 16,715, respectively. Consequently, the prevalence estimate is 12.29%, which is close to 10%. However, it is crucial to acknowledge that we cannot assert that Taiwan Biobank participants are entirely representative of the general population in Taiwan. Volunteer-based studies, including those conducted in the Taiwan Biobank and the UK Biobank (Fry et al., 2017; Keyes & Westreich, 2019), often exhibit sample distributions that may not perfectly mirror the overall population. Although the data may not be suitable for deriving universally applicable disease prevalence and incidence rates, the substantial sample size of the biobanks and the diversity in exposure measures contribute to valid scientific inferences regarding associations between exposures and health conditions applicable to broader populations (Fry et al., 2017).

14. Need to clarify how sample for training or validation was determined.

Response:

In this study, the allocation of training, validation, and testing data was conducted through random partitions. Specifically, for the genetic-centric analysis, samples containing only baseline data (without follow-up) were randomly divided into training and validation sets at an 8:2 ratio, while samples with follow-up data were designated as the testing set. We have added the detailed description in Lines 111 – 117 in the Methods section in the revised manuscript as follows: “We assigned 80% and 20% of data as the training and validation samples. The GWAS samples (Dataset 1, N = 35,688), training samples (Dataset 2, N = 12,236; Dataset 4, N = 40,787), and validation samples (Dataset 3, N = 3,060; Dataset 5, N = 10,197). For classification analysis, testing samples comprised two independent datasets: Dataset 6 (N = 8,827) and Dataset 7 (N = 936), while for prediction analysis, they were represented as Datasets 6' (N = 8,827) and Dataset 7' (N = 936) (Fig. 1A and Fig. S1).”

In the genetic-imaging integrative analysis, all samples, which included both baseline and follow-up data, underwent a two-step partitioning process. Initially, the entire dataset was split into training + validation and testing sets at an 8:2 ratio. Subsequently, the training + validation set was further randomized into distinct

training and validation datasets, maintaining an 8:2 ratio. We have added the above description in Lines 160 – 163 in the Methods section in the revised manuscript. This approach was adopted to ensure the robustness of our models and the generalizability of the results across different datasets.

15. Lines 358-366. These sentences simply repeated the same thing twice!

Response:

In response to the reviewer's comment on Lines 358 – 366 in the original manuscript, we have rephrased the corresponding description in Lines 471 – 480 in the revised manuscript as follows: "In this study, we conducted a comparative analysis of two prediction models based on GWAS data. The first method utilized T2D-associated SNPs derived from our GWAS with a limited sample size, either as individual predictors or in combination to construct a PRS for the prediction model. The second method incorporated T2D-associated SNPs from previously published GWASs with a significantly larger sample size or utilized summary statistics of whole-genome SNPs from GWASs with a considerably larger sample size to construct the PRS. Notably, the latter approach yielded a higher prediction AUC. These findings underscore the substantial impact of sample size in GWAS, PRS construction, and subsequent classification and prediction analyses, aligning with previous research (Wray et al., 2008)."

16. The study participants and variables section may be more appropriate to be placed within method section.

Response:

We have incorporated the reviewer's suggestion and relocated the study participants and variables section to Lines 104 – 169 in the Method section, aligning with the recommended structure.

Reviewer #2 (Remarks to the Author):

Risk assessment and early detection of T2D are vital in improving individuals' health and in this study, authors established a prediction model and identified high-risk subgroups for T2D with 68,911 Taiwan Biobank (TWB) participants. Authors demonstrate that genetic information alone is insufficient for accurate T2D prediction (AUC = 0.73), and incorporating medical imaging data significantly improves prediction accuracy (AUC = 0.89). The best-performing model integrates genetic, medical imaging, and demographic variables (AUC = 0.94). The study also presents an online risk assessment website for T2D. Manuscript comprehensively evaluated their prediction model. Here are my comments.

Response:

Thank you very much for the summary and comments on our work. We address the comments raised by this reviewer below.

1. In this manuscript, authors used GWAS based on 50,984 subjects. There have been multiple GWAS studies using more than 100,000 east asian (for instance, Spracklen et al, Nature, 2000). It seems better to use those larger GWAS summary statistics when authors built PRS and any comparison is required.

Response:

We fully agree with the reviewer's suggestion regarding the utilization of larger GWAS summary statistics when constructing PRS. In fact, this precisely aligns with our approach in this study. Our PRS was developed using extensive GWAS summary statistics from the DIAGRAM Consortium, which includes data from over one million participants.

For additional clarity, we have moved the description from **Supplemental Text 1** in the original manuscript to the main text in Lines 171 – 187 in the Methods section in the revised manuscript as follows: "PRS was constructed by using PRS-CS (Ge et al., 2019) and PRS-CSx (Ruan et al., 2022). PRS-CS (Ge et al., 2019) based on the meta-GWAS summary statistics of T2D in East Asia in the DIAGRAM Consortium (Mahajan et al., 2022) and the linkage disequilibrium (LD) reference from the EAS population of the 1000 Genomes Project (Auton et al., 2015). PRS was calculated using PLINK (--score command) based on our genotype data, and 884,327 SNP effects were estimated using PRS-CS. Normalized PRS was standardized to mean = 0 and standard deviation = 1. PRS-CSx (Ruan et al., 2022) based on the meta-GWAS summary statistics of T2D in multiple populations, including (a) East Asian: 56,268 cases and 227,155 controls in the DIAGRAM Consortium (Mahajan et al., 2022); (b) European: 80,154 cases and 853,816

controls in the DIAGRAM Consortium (Mahajan et al., 2022); (c) South Asian: 16,540 cases and 32,952 controls in the DIAGRAM Consortium (Mahajan et al., 2022), and the LD reference from each of the three populations (EAS, EUR, and SAS). 884,327, 880,098, and 900,047 SNPs for EAS, EUR, and SAS were applied to our data to calculate the population-specific PRS for each individual using the PLINK (--score command). We combined the three population-specific PRS with equal weight to calculate a final PRS. R language was used to standardize the PRS to mean = 0 and standard deviation = 1.”

Furthermore, we provide additional context in Lines 471 – 480 in the Discussions section in the revised manuscript: “In this study, we conducted a comparative analysis of two prediction models based on GWAS data. The first method utilized T2D-associated SNPs derived from our GWAS with a limited sample size, either as individual predictors or in combination to construct a PRS for the prediction model. The second method incorporated T2D-associated SNPs from previously published GWASs with a significantly larger sample size or utilized summary statistics of whole-genome SNPs from GWASs with a considerably larger sample size to construct the PRS. Notably, the latter approach yielded a higher prediction AUC. These findings underscore the substantial impact of sample size in GWAS, PRS construction, and subsequent classification and prediction analyses, aligning with previous research (Wray et al., 2008). Consequently, in situations where the sample size is limited, we propose utilizing external genetic information such as SNPs and summary statistics from published studies with larger sample sizes, which not only facilitates the development of a more predictive PRS and model but also reduces computational overhead (Brown et al., 2016).”

2. Authors showed that prediction model including age, sex, family history of T2D, and PRS achieved AUC=0.915. This result is very interesting. It would be interesting to evaluate the importance of each variable. For instance, AUCs from the model with age and without age can be compared and their improvement can be utilized as the variable importance.

Response:

Thank you for the constructive comment. We have assessed the importance of each variable in our prediction model and incorporated the results in Lines 284 – 296 in the Results section in the revised manuscript as follows: “The importance of each predictor was evaluated through a backward elimination procedure of variables. The optimal model incorporating age, sex, family history of T2D, and PRS achieved an AUC of 0.915. The AUC reductions upon removing individual variables are as follows: (a) Omitting the age variable resulted in an AUC of 0.839, representing a reduction of 0.076. (b) Excluding the sex variable resulted in an AUC of 0.905, with a reduction of

0.01. (c) Removing the family history of T2D variable yielded an AUC of 0.881, with a reduction of 0.034. (d) Eliminating the PRS variable resulted in an AUC of 0.884, with a reduction of 0.031. Based on the reduction in AUC, the impact size appears to be in the order of: age > family history > PRS > sex. Additionally, we evaluated feature importance using a feature importance score, calculated based on the Gini index (Breiman et al., 1984) (see the **Methods** section). According to this analysis, the effect size is: family history > age > PRS > sex. In summary, our findings consistently highlight age and family history as the most crucial risk factors for T2D.”

3. According to the AUC, the inclusion of age already generates the very large AUC (0.804). It seems that the T2D patients are generally very old, and because of this issue, the prediction accuracy can become very large. If it is true, then I think that prediction accuracy must be evaluated without age.

Response:

In the Taiwan Biobank (TWB) data, T2D patients tend to be older than non-T2D participants, as indicated in **Table A1** in this reply letter (the new **Table 1** in the revised manuscript). Based on T2D Definition IV, the optimal model, incorporating age, sex, family history of T2D, and PRS, achieves an AUC of 0.9147, Accuracy of 0.843, Sensitivity of 0.844, Specificity of 0.843, and F1 of 0.672. Notably, our testing dataset demonstrates effective prediction for both T2D cases in the aged group (>65 years old) and the non-aged group. In the aged group, with 64 T2D patients, the accuracy is 0.93, while in the non-aged group (275 young T2D patients), the accuracy remains reasonably high at 0.82.

Regarding to the reviewer’s suggestion, we conducted an additional analysis excluding the influential variable, age, as depicted in **Fig. 3D** in the original manuscript. The model, comprising sex, family history of T2D, and PRS, yields an AUC of 0.8388, Accuracy of 0.7466, Sensitivity of 0.7552, Specificity of 0.7446, and F1 of 0.5322. The performance metrics show a decrease compared to the best model that includes age. In the aged group, with 64 T2D patients, the accuracy is 0.75, and in the non-aged group (275 young T2D patients), the accuracy is 0.75 as well. This reduction in accuracy after excluding age underscores the crucial role of age as a predictor for T2D.

Table A1. Demographic characteristics of the study population for four T2D definitions. Sample size, mean and median age with standard deviation (SD), the number and proportion of males, and their differences in the T2D case and control groups are provided. P-values for two-sample two-sided t test and chi-square test are displayed. Note that the total sample size in this table is N = 59,811, comprising 40,787 participants in the training dataset, 10,197 participants in the validation dataset, and 8,827 participants in the first testing dataset (refer to **Fig. 1**). Note that the participants in the second independent testing dataset (N = 936) (refer to Datasets 7 and 7' in **Fig. 1**) are only used for replication purposes and are therefore not included in this table.

Phenotype		Overall	T2D group	Non-T2D group	
Def I	N	59,811	3,011	56,800	
	Age, mean (SD) (years)	50.57 (10.57)	58.30 (7.62)	50.16 (10.54)	$p \approx 0$
	Age, median (years)	52	60	51	
	Sex, male, n (%)	18,973 (31.72%)	1,350 (44.83%)	17,623 (31.02%)	$p = 1.48 \times 10^{-56}$
Def II	N	30,227	2,320	27,907	
	Age, mean (SD) (years)	47.87 (10.71)	58.22 (7.66)	47.01 (10.48)	$p \approx 0$
	Age, median (years)	48	59	46	
	Sex, male, n (%)	8,962 (29.64%)	1,055 (45.47%)	7,907 (28.33%)	$p = 2.11 \times 10^{-67}$
Def III	N	32,520	1,155	31,365	
	Age, mean (SD) (years)	49.25 (10.67)	58.28 (7.55)	48.91 (10.62)	$p = 1.36 \times 10^{-235}$
	Age, median (years)	50	59	50	
	Sex, male, n (%)	8,987 (27.63%)	537 (46.49%)	8,450 (26.94%)	$p = 5.07 \times 10^{-48}$
Def IV	N	20,545	2,393	18,152	
	Age, mean (SD) (years)	47.95 (10.84)	58.24 (7.63)	46.59 (10.47)	$p \approx 0$
	Age, median (years)	48	59	46	
	Sex, male, n (%)	5,746 (27.96%)	1,091 (45.59%)	4,655 (25.64%)	$p = 1.36 \times 10^{-92}$

4. External validation is necessary. For instance, authors can apply to subjects in different cities or the population in the other east asian country.

Response:

We appreciate the reviewer's insightful suggestion for external validation, and we acknowledge the importance of further validation with an external dataset. Integrative studies involving genetic and multi-modality medical imaging variables for disease risk evaluation are at the forefront of research and necessitate time to accumulate sufficient data.

While we agree that external validation with a diverse dataset would be ideal, it is essential to recognize the current limitations in the availability of publicly accessible datasets that simultaneously include genetic data and all four types of medical imaging data, especially for Asian populations. Our study, uniquely positioned within the Taiwan Biobank, fills a critical gap by providing a comprehensive analysis for the

Asian population, contributing significantly to the scarcity of Asian data in global research. The distinctive nature of our dataset, not readily available in other biobanks or initiatives like the UK Biobank, underscores the pioneering nature of our study.

We believe that our proof-of-principle study, though conducted within the Taiwan Biobank, is forward-looking and emphasizes the promising trend of combining genetic and medical imaging data for future disease risk evaluation. While we acknowledge the evolving landscape and anticipate the gradual emergence of datasets with similar characteristics, waiting for their availability could substantially delay critical discoveries. In our discussion, we mention the potential for datasets of this nature to become available gradually, with self-funded health checkups being a potential source for such tests.

To enhance the robustness of our findings, we have rigorously replicated our model using two independent testing datasets within the Taiwan Biobank, as extensively detailed in the paper. Additionally, we are actively pursuing collaboration with a medical center, seeking Institutional Review Board (IRB) approval to collect external validation data. However, this process is time-consuming, and we anticipate delays in obtaining and incorporating external datasets for future validation.

We have incorporated a new paragraph into the Discussion section of the revised manuscript in Lines 572 – 579 to highlight the significance of external validation for our findings as follows: “While our current study presents a robust proof-of-principle model for disease risk evaluation based on genetic and multi-modality medical imaging variables within the Taiwan Biobank, we recognize the importance of external validation for broader generalizability. However, the current landscape poses challenges in accessing publicly available datasets that encompass both genetic and all four types of medical imaging data. Despite the inherent limitations in readily available datasets with comparable characteristics, we are actively engaged in collaborations with a medical center to collect external validation data for future studies.”

We sincerely appreciate the reviewer’s understanding of the complexities involved in accessing external datasets and our commitment to advancing precision medicine. We are dedicated to further validating our model and will actively work towards incorporating external datasets in subsequent studies. Your feedback is invaluable in strengthening the integrity and applicability of our work, and we are eager to address any additional concerns or suggestions you may have.

5. Authors considered four different definitions of T2D. In the final results, authors found that T2D definition IV has the highest accuracy, and it was applied to the Figure

5 and 6. I think that it is better to consider the model with better diagnostic accuracy because the prediction accuracy depends on the characteristics of the selected cohorts.

Response:

We appreciate the reviewer's observation and agree with the suggestion. Among the four T2D definitions considered, we focused on T2D Definition IV in this study, specifically self-reported T2D + HbA1c + GLU-AC, as it aligns most closely with clinical practice. Our results also show that the model based on the T2D Definition IV has the highest prediction accuracy. Therefore, in this study, better diagnostic accuracy and higher prediction accuracy are actually consistent.

We have added a new paragraph in Lines 485 – 496 in the Discussion section in the revised manuscript as follows: "In our study, we investigated the importance of employing a precision phenotype definition for the evaluation of disease risk. We also addressed a potential limitation associated with the prevalent use of self-reported disease status. Utilizing four T2D definitions, Type IV, which integrates self-reported T2D with measurements of HbA1c and GLU-AC, emerged as a definition closely aligned with clinical practice. Our results demonstrate that the model based on T2D Definition IV exhibits the highest prediction accuracy, despite the presence of some missing data for HbA1c or GLU-AC. Consequently, in this study, superior diagnostic accuracy corresponds with higher prediction accuracy. Furthermore, the application of self-reported T2D (Definition Type I) yields an AUC significantly lower than the AUC of Definition Type IV. This outcome underscores a potential limitation associated with the commonly used self-report disease status, which functions as a convenient phenotype in the analysis of Taiwan Biobank data."

6. It seems that the data that authors used may include more older subjects. The prediction model and its accuracy must be affected by the sampling error. Authors should provide how to handle this issue.

Response:

In response to the concern regarding a potential over-representation of older subjects in our study, we conducted a comprehensive analysis of the Taiwan Biobank data. The proportion of older subjects in the Taiwan Biobank is slightly lower than (NOT higher than) that observed in the general population of Taiwan. To provide transparency, we have included an annual breakdown of participant recruitment since 2008, categorized into an older subject group (>65 years old) and a non-aged group (<64 years old) (refer to **Fig. A4** in this response letter).

When comparing our findings to the demographic survey data from the Ministry of the Interior, Taiwan (<https://www.ris.gov.tw/app/portal/346>), the proportion of the older subject group in the Taiwan Biobank is 8.9%, while in the entire Taiwan

population, it is 13.09%. Notably, Taiwan transitioned to an aging society in 1993, became an aged society in 2018, and is estimated to enter a super-aged society by 2025 (https://www.ndc.gov.tw/Content_List.aspx?n=D527207EEEF59B9B).

It is important to acknowledge that volunteer-based studies, such as those conducted in the Taiwan Biobank and the UK Biobank (Fry et al., 2017; Keyes & Westreich, 2019), often exhibit sample distributions that may not perfectly mirror the overall population. While the data may be not suitable for deriving universally applicable disease prevalence and incidence rates, the substantial sample size of the biobanks and the diversity in exposure measures contribute to valid scientific inferences regarding associations between exposures and health conditions that are applicable to broader populations (Fry et al., 2017).

Figure A4. Demographic survey data from the Ministry of the Interior, Taiwan. Source of the data is from the website (<https://www.ris.gov.tw/app/portal/346>).

Minor comments:

1. For https://hcyang.stat.sinica.edu.tw/software/T2D_web/header.php, authors provide the percentage of PRS as a guideline. For instance, 0-10% is -3.488 - -1.28. However I think that those information is not practical because those depends on the number of missing SNP, and the scale, etc.

Response:

Thank you for bringing this to our attention. We agree with the reviewer's observation

that the comparability of PRS can be influenced by factors such as the number of missing SNPs and the scale used.

Regarding the concern about missing SNPs, during PRS calculation, we utilized data from whole-genome SNP imputation. If the proportion of missing SNPs is not high, its impact on the PRS should be limited.

To address the potential impact of scale on PRS, we implemented the following measures: initially, we normalized the PRS by standardizing it to have a mean of 0 and a standard deviation of 1. Subsequently, we rescaled it to a percentage, departing from the use of the original PRS scores. This transformation allows users to better understand the distribution characteristics within our dataset. To enhance the transparency of our study, we have also added a new **Table 1** in the revised manuscript detailing the demographic distribution in our revised manuscript. This additional information facilitates a clearer understanding of the study population.

However, we acknowledge that the current results may not be flawless. PRS comparability between different studies or datasets is indeed challenging due to various factors, including differences in genotyping platforms, imputation methods, SNP inclusion criteria, and population characteristics. Additional adjustments may be necessary in real-world applications.

It's important to emphasize that the comparability of PRS across studies is an ongoing challenge in the field of genetics. Researchers should exercise caution when making direct comparisons and consider providing detailed information about the methods used. Exploring strategies to standardize or harmonize PRS values can contribute to more meaningful cross-study comparisons.

We value your insightful observation and will exercise caution in implementing and applying the tool. Should the reviewer persist in recommending the removal of our online risk calculator from this paper, we will follow the reviewer's suggestion.

2. I think that authors should provide the SNPs that they used for calculating PRS and the detailed information about how to calculate PRS.

Response:

Thank you very much for the suggestion. We provide an example file for the T2D risk calculator. In the file, all of SNPs used for calculating PRS in this study are provided.

Regarding the construction of PRS, we have added the description regarding PRS construction in Lines 171 – 187 in the Methods section in the revised manuscript as follows: "PRS was constructed by using PRS-CS (Ge et al., 2019) and PRS-CSx (Ruan et al., 2022). PRS-CS (Ge et al., 2019) based on the meta-GWAS summary statistics of T2D in East Asia in the DIAGRAM Consortium (Mahajan et al., 2022) and the linkage disequilibrium (LD) reference from the EAS population of the 1000 Genomes Project

(Auton et al., 2015). PRS was calculated using PLINK (--score command) based on our genotype data, and 884,327 SNP effects were estimated using PRS-CS. Normalized PRS was standardized to mean = 0 and standard deviation = 1. PRS-CSx (Ruan et al., 2022) based on the meta-GWAS summary statistics of T2D in multiple populations, including (a) East Asian: 56,268 cases and 227,155 controls in the DIAGRAM Consortium (Mahajan et al., 2022); (b) European: 80,154 cases and 853,816 controls in the DIAGRAM Consortium (Mahajan et al., 2022); (c) South Asian: 16,540 cases and 32,952 controls in the DIAGRAM Consortium (Mahajan et al., 2022), and the LD reference from each of the three populations (EAS, EUR, and SAS). 884,327, 880,098, and 900,047 SNPs for EAS, EUR, and SAS were applied to our data to calculate the population-specific PRS for each individual using the PLINK (--score command). We combined the three population-specific PRS with equal weight to calculate a final PRS. R language was used to standardize the PRS to mean = 0 and standard deviation = 1.”

Reviewer #3 (Remarks to the Author):

This is a thoughtfully done analysis of genetic, clinical, and demographic risk factors for type 2 diabetes (T2D). It is the first such analysis carried out using a unique and valuable data resource from Taiwan. The work is heavily predictive in focus, and does not question or propose mechanisms relating to the disease. Nevertheless, such predictive analyses are valuable in helping to characterize the genetic contribution to T2D risk. The implications of the short follow-up imaging study are less clear. I appreciate that the authors acknowledge the potential for misclassification of T2D status and conduct some investigations indicating that high-risk people classified as controls may in some sense be pre-diabetic.

Response:

Thank you very much for the summary and comments on our work. We address the comments raised by this reviewer below.

Line 31 in the abstract states that the prediction involving genetics, imaging, and demographics identifies subjects at high risk of developing T2D (AUC is a remarkable 0.94). However given the timing of the imaging analysis and T2D diagnosis, it is unclear if this is actually predicting future risk or current T2D status.

Response:

We appreciate the reviewer's insightful comment regarding the timing of medical imaging data availability in the Taiwan Biobank. It is acknowledged that the imaging data were only accessible during the follow-up period, which is a limitation of our current study. While we recognize the importance of distinguishing between predicting future risk and current T2D status, the focus of our analysis was on establishing a stable and accurate classification model using available medical imaging data. We hope that this model has the potential to serve as a reliable surrogate for predicting future disease risk.

As discussed in the manuscript's Discussion section, the Taiwan Biobank continues to follow up with participants, and we anticipate the accrual of additional data in the future. This ongoing data collection will allow us to validate and refine our results with the inclusion of new information.

Specific comments:

1. The cohort is insufficiently characterized, e.g. I could not even find the median age and sex distribution. Is the cohort in any sense representative of the Taiwan population?

Are the cases and controls in any sense comparable with respect to age? I assume that the cohort is overwhelmingly Han Chinese but this information should be provided.

Response:

Thank you for your valuable feedback. In response to your suggestion, we have included a new **Table 1** in the revised manuscript (see **Table A1** in this reply letter), providing sample size, mean and median age with standard deviation (SD), the number and proportion of males, and their differences in the T2D case and control groups. The table reveals that the participants in the T2D case group are older than those in the control group, with a higher proportion of males.

Regarding the age distribution, we conducted an additional analysis of the Taiwan Biobank data. The proportion of older subjects in the Taiwan Biobank is slightly lower than that observed in the general population of Taiwan. To provide transparency, we have included an annual breakdown of participant recruitment since 2008, categorized into an older subject group (>65 years old) and a non-aged group (<64 years old) (refer to **Fig. A4** in this response letter).

When comparing our findings to the demographic survey data from the Ministry of the Interior, Taiwan (<https://www.ris.gov.tw/app/portal/346>), the proportion of the older subject group in the Taiwan Biobank is 8.9%, while in the entire Taiwan population, it is 13.09%. Notably, Taiwan transitioned to an aging society in 1993, became an aged society in 2018, and is estimated to enter a super-aged society by 2025 (https://www.ndc.gov.tw/Content_List.aspx?n=D527207EEEF59B9B).

It is important to acknowledge that volunteer-based studies, such as those conducted in the Taiwan Biobank and the UK Biobank (Fry et al., 2017; Keyes & Westreich, 2019), often exhibit sample distributions that may not perfectly mirror the overall population. While the data may be not suitable for deriving universally applicable disease prevalence and incidence rates, the substantial sample size of the biobanks and the diversity in exposure measures contribute to valid scientific inferences regarding associations between exposures and health conditions that are applicable to broader populations (Fry et al., 2017).

In addition, we have included information about the ethnicity of our study population in Line 89 in the Introduction section in the revised manuscript as follows: "The TWB enrolled participants aged over 20 from the Han Chinese population in Taiwan ..."

Table A1. Demographic characteristics of the study population for four phenotype definition. Sample size, mean and median age with standard deviation (SD), the number and proportion of males, and their differences in the T2D case and control groups are provided. P-values for two-sample two-sided t test and chi-square test are displayed. Note that the total sample size in this table is N = 59,811, comprising 40,787 participants in the training dataset, 10,197 participants in the validation dataset, and 8,827 participants in the first testing dataset (refer to **Fig. 1**). Note that the participants in the second independent testing dataset (N = 936) (refer to Datasets 7 and 7' in **Fig. 1**) are only used for replication purposes and are therefore not included in this table.

Phenotype		Overall	T2D group	Non-T2D group	
Def I	N	59,811	3,011	56,800	
	Age, mean (SD) (years)	50.57 (10.57)	58.30 (7.62)	50.16 (10.54)	$p \approx 0$
	Age, median (years)	52	60	51	
	Sex, male, n (%)	18,973 (31.72%)	1,350 (44.83%)	17,623 (31.02%)	$p = 1.48 \times 10^{-56}$
Def II	N	30,227	2,320	27,907	
	Age, mean (SD) (years)	47.87 (10.71)	58.22 (7.66)	47.01 (10.48)	$p \approx 0$
	Age, median (years)	48	59	46	
	Sex, male, n (%)	8,962 (29.64%)	1,055 (45.47%)	7,907 (28.33%)	$p = 2.11 \times 10^{-67}$
Def III	N	32,520	1,155	31,365	
	Age, mean (SD) (years)	49.25 (10.67)	58.28 (7.55)	48.91 (10.62)	$p = 1.36 \times 10^{-235}$
	Age, median (years)	50	59	50	
	Sex, male, n (%)	8,987 (27.63%)	537 (46.49%)	8,450 (26.94%)	$p = 5.07 \times 10^{-48}$
Def IV	N	20,545	2,393	18,152	
	Age, mean (SD) (years)	47.95 (10.84)	58.24 (7.63)	46.59 (10.47)	$p \approx 0$
	Age, median (years)	48	59	46	
	Sex, male, n (%)	5,746 (27.96%)	1,091 (45.59%)	4,655 (25.64%)	$p = 1.36 \times 10^{-92}$

Figure A4. Demographic survey data from the Ministry of the Interior, Taiwan. Source of the data is from the website (<https://www.ris.gov.tw/app/portal/346>).

2. The paper lacks context in relation to the known epidemiologic characteristics of type 2 diabetes, even with respect to basic factors such as age and sex differences. Surprisingly there is no mention of BMI in the paper.

Response:

In response to the reviewer’s feedback regarding the contextualization of our study in relation to known epidemiologic characteristics of Type 2 Diabetes (T2D), particularly age and sex differences, we have included a new **Table 1** in the revised manuscript (see **Table A1** in this reply letter), detailing the characteristics of our study participants in the revised manuscript. This table provides information on the mean and median age and sex distribution, and their differences in the T2D case and control groups. The table reveals that the participants in the T2D case group are older than those in the control group, with a higher proportion of males.

Regarding the absence of body mass index (BMI) as a risk predictor in our study, we acknowledge its importance in T2D research. However, we deliberately excluded BMI from our analysis due to its susceptibility to external factors such as food preferences and behavior, making it a less stable biomarker for T2D and metabolic disorders (Matafome (2020); Strings et al. (2023); Timothy Garvey (2019)). Our decision aligns with the recognition that traditional glycemc measurements, including HbA1c and fasting glucose, have demonstrated limitations in disease risk prediction in various studies (Buchanan et al., 2009; Eehalt et al., 2017; Greenhalgh et al., 2011; John, 2022; Li et al., 2018; Nowicka et al., 2011; Panwar et al., 2013; Spiller et al., 2018).

These metrics often show limited sensitivity, especially in specific populations, and lack accuracy in predicting pre-diabetes and T2D. Of importance, our study focuses on precision medicine, aiming to identify high-risk subgroups rather than adopting a one-size-fits-all approach. Leveraging genetic and medical imaging data provides more detailed insights into individual characteristics, enabling personalized risk assessment.

3. I'm not sure how to interpret figure 5b. T2D status at the follow-up time point is an outcome, so it's not clear what is the meaning of stratifying the estimated survival functions on a future event.

Response:

In the genetic-centric analysis, we established the prediction model to predict participants' T2D status at follow-up based on the data at baseline. Therefore, we have observed T2D status at baseline, observed T2D status at follow-up, and predicted T2D status at follow-up. In the analysis of **Fig. 5** in the original manuscript, we focused on the participants who were non-T2D (i.e., control) at baseline.

For these participants who were non-T2D at baseline and became T2D at follow-up, if they were predicted as a case by our prediction model, then they belonged to G1 group; if they were predicted as a control by our prediction model, then they belonged to G2 group. For the participants who were non-T2D at baseline and follow-up, if they were predicted as a case by our prediction model, then they belonged to G3 group; if they were predicted as a control by our prediction model, then they belonged to G4 group.

Regarding T2D event time (i.e., survival time), it's defined as follows: At follow-up, some of these participants became a T2D case and some remained as a control. For the participants who were non-T2D at baseline and became T2D at follow-up, the event time (i.e., survival time) was calculated by the time episode between the baseline and follow-up. For the participants who were non-T2D at baseline and follow-up, the event time is censored. Finally, we analyzed the event time of the four groups (G1, G2, G3, and G4) by using Cox regression.

The major results are summarized in the original and revised manuscripts as follows: First, compared to G4 ("true negative"), G3 had a significantly lower T2D-free probability (**Fig. 5B**), shorter median survival time (**Fig. 5C**), higher T2D-risk under similar follow-up time (**Fig. 5D** and **Fig. 5E**), higher HbA1c (**Fig. 5F**), and higher fasting glucose (**Fig. 5G**). Second, compared to G1 ("true positive"), G3 had a comparable survival rate (**Fig. 5B**), median survival time (**Fig. 5C**), and T2D-risk under similar follow-up time (**Fig. 5D** and **Fig. 5E**) but lower HbA1c (**Fig. 5F**) and fasting glucose (**Fig. 5G**).

4. Due to the strong age-dependent nature of T2D onset, an alternative to including baseline age as a covariate in the Cox model is to use age as the time variable with age at entry as a left truncation time. Then the baseline hazard would capture age-specific risk. Stratifying by sex, rather than including sex as a covariate is also something to consider. The HR for sex (50% lower risk for females) is remarkably strong and warrants discussion.

Response:

Thank you for your valuable feedback. First, in response to the reviewer's suggestion, our additional analysis comprehensively considered two types of time scales (i.e., time-on-study and age) and three types of sex variable treatment (i.e., adjusting for sex as a covariate, conducting sex-specific analysis with an assumption of a common sex effect, and performing sex-specific analysis with different sex effects), resulting in six analyses (refer to **Table A2** in this reply letter and **Supplementary Table S5** in the revised manuscript). The results of the six analyses are consistent and indicate that increased age, higher PRS, and a stronger family history of T2D are associated with a higher risk of T2D. Specifically, elderly males with a strong family history of T2D and high PRS exhibit a particularly severe risk of developing T2D.

We have added the description on Lines 221 – 236 in the Method section "...The Cox regression analysis considered two types of time scales (i.e., time-on-study and age) and three types of sex variable treatment (i.e., adjusting for sex as a covariate, conducting sex-specific analysis with an assumption of a common sex effect, and performing sex-specific analysis with different sex effects), resulting in six analyses (refer to **Supplementary Table S5**).

The initial three analyses considered time-on-study as the time scale, with age at baseline included as a covariate, and incorporating the following sex variable treatment: (1) Model 1: Sex was treated as a covariate in the analysis; (2) Model 2: Sex-specific analysis, assuming a common effect for males and females; (3) Model 3: Sex-specific analysis with different effects for males and females. The subsequent three analyses considered age as the time scale, with age-at-baseline as left truncation, along with the three sex variable treatments, similar to the time-on-study analysis, to be Models 4 – 6.

In the time-on-study scale analysis, the event time was calculated as the duration from baseline to follow-up. In the age scale analysis, the event time was age-at-follow-up. In each model, ..."

We have also added the description on Lines 347 – 356 in the Results section "...The Cox regression analyses that considered two types of time scales and three types of sex variable treatment obtained a consistent result (**Supplementary Table S5**). Using the analysis in which we considered time-on-study as the time-scale with age at

baseline, sex, family history of T2D, and PRS as covariates for illustration, age, sex, family history of T2D, and PRS were all significantly associated with T2D ($p < 0.001$) (**Fig. 4D**). Increased age, higher PRS, and stronger T2D family history had a higher T2D risk. The elderly male, with a strong family history and high PRS, had a severe T2D risk (**Fig. 4E** for multivariate Cox regression and **Fig. S8** for univariate Cox regression). We also provided the predicted time-to-event (week) (**Fig. 4F**) ...”

Second, regarding the reviewer’s concern about the remarkably strong hazard ratio (HR) for female compared to sex (50% lower risk for females); female HR compare to male = 0.495). We found that this strong effect was also reported in other reference. For instance, in Lin et al. (2020), **Table A2** in the response letter (**Supplementary Table S5** in the revised manuscript) the study reported a female HR = 0.427, close to our result.

Table A2. Cox regression analysis with different considerations of time scales and sex variable treatments. The Cox regression analysis considered two types of time scales (i.e., time-on-study and age) and three types of sex variable treatment (i.e., adjusting for sex as a covariate, conducting sex-specific analysis with an assumption of a common sex effect, and performing sex-specific analysis with different sex effects), resulting in six analyses. The initial three analyses considered time-on-study as the time scale, with age at baseline included as a covariate, and incorporating the following sex variable treatment: (1) Model 1: Sex was treated as a covariate in the analysis; (2) Model 2: Sex-specific analysis, assuming a common effect for males and females; (3) Model 3: Sex-specific analysis with different effects for males and females. The subsequent three analyses considered age as the time scale, with age-at-baseline as left truncation, along with the three sex variable treatments, similar to the time-on-study analysis, to be Models 4 – 6.

Time scale: Time-on-study				
Covariates	Model 1	Model 2	Model 3	
	Sex as covariate	Sex-specific analysis, assuming a common effect for males and females	Sex-specific analysis with different effects for males and females	
HR (95% CI)				
			Male	Female
Age	1.087 (1.067, 1.106)	1.087 (1.067, 1.106)	1.057 (1.027,1.088)	1.103 (1.078,1.129)
Sex-Female	0.557 (0.419,0.741)	-	-	-
Family	1.640 (1.410, 1.907)	1.636 (1.407, 1.903)	1.534 (1.146,2.053)	1.666 (1.394,1.991)
PRS	1.643 (1.436, 1.880)	1.651 (1.442, 1.890)	1.840 (1.458, 2.322)	1.565 (1.325,1.848)

(continued).

Time scale: Age				
Covariates	Model 4	Model 5	Model 6	
	Sex as covariate	Sex-specific analysis, assuming a common effect for males and females	Sex-specific analysis with different effects for males and females	
HR (95% CI)				
			Male	Female
Age	-	-	-	-
Sex-Female	0.582 (0.439, 0.773)	-	-	-
Family	1.585 (1.368, 1.836)	1.568 (1.353,1.817)	1.521 (1.140,2.029)	1.591 (1.339, 1.890)
PRS	1.659 (1.448, 1.900)	1.661 (1.449,1.904)	1.765 (1.398,2.230)	1.608 (1.359, 1.903)

5. *The analysis of exercise is limited. At most 290 people report any specific type of exercise. The denominator for these numbers is unclear but it seems to me that exercise may be under-reported in this cohort. The lack of clear results may be due to low power. Testing for multiple types of exercise seems not to have been subject to rigorous multiplicity adjustment.*

Response:

In the original manuscript, we explored the impact of exercise on HbA1c levels and the risk of T2D using only the samples from the testing dataset, resulting in a limited sample size of N = 1,776. However, recognizing the potential under-reporting of exercise in this cohort and understanding that the evaluation of exercise’s association

with HbA1c and T2D risk is exploratory and not predictive, we revised our approach. The analysis was conducted using the entire study cohort, substantially increasing the sample size to N = 59,811.

Moreover, to address concerns of potential Type I errors due to multiple testing in our analysis of various types of exercise and phenotypes, we applied Bonferroni's correction, adjusting the significance level to $0.05/(22 \times 5) = 4.5 \times 10^{-4}$. The results are presented in **Fig. S9** (see **Fig. A5** in the response letter), demonstrating the continued benefits of walking for fitness. We acknowledge the need for future intervention studies to validate these findings.

We have revised the description in Lines 361 – 375 in the Results section in the revised manuscript as follows: “To assess the impact of exercise on HbA1c, linear regression analysis was performed. Multiple testing for 110 analyses was corrected using Bonferroni correction, and the level of significance was set as 4.5×10^{-4} . It was observed that individuals engaging in regular exercise experienced a significant reduction in HbA1c by an average of 0.09% mg/dL ($p < 0.001$) compared to those who did not engage in regular exercise. Moreover, individuals with a high PRS who engaged in exercise demonstrated a greater reduction in HbA1c (0.13% mg/dL) than those with a low PRS (0.08% mg/dL). Additionally, the results also suggested that the T2D patients who regularly engaged in exercise can have a noteworthy improvement of 0.32% mg/dL in HbA1c than those T2D patients who did not do regular exercise. In addition, among the various types of exercise, walking for fitness exhibited the most robust reduction in HbA1c for all samples, including high and low-risk subgroups, and both T2D and non-T2D groups (**Fig. S9**). On average, participants engaged in walking for fitness 18.30 times a month (standard deviation = 8.64) for approximately 48.13 minutes per session (standard deviation = 22.92).”

Figure A5. Effect of doing exercise on HbA1c. Beta coefficients (B) for various types of exercise are displayed via a bar chart. In total 59,811 samples, the figures on the right-hand side indicate the number of individuals for doing each type of exercise. Only the result of an analysis containing a sample size > 30 is shown. **(A) All samples; (B) PRS.** Beta coefficients for the high-risk group (red bar) and non-high-risk group (blue) are displayed; **(C) T2D groups.** Beta coefficients for the T2D group (red bar) and non-T2D group (blue) are displayed. Walking for fitness was found to be significantly negatively associated with T2D in all samples, including high and low-risk subgroups, and both T2D and non-T2D groups.

6. It would be helpful to consider how the PRS and familial T2D status are related. This may shed light on the contributions of genetics, epigenetics, and shared environment.

Response:

We appreciate the reviewer’s insightful suggestion regarding the relationship between PRS and familial T2D status. Family history is encompassing genetics, epigenetics, and shared environment. In response, we conducted additional analyses in the revised manuscript to explore these connections more thoroughly. We have added a paragraph entitled “Assessment of family history of T2D” into the Results section in Lines 298 – 318 in the revised manuscript as follows: “Family history is encompassing genetics and environment. We delved into the connection between the family history of T2D – treated as a graded-scale (0, 1, 2, 3, and 4) – and the genetic component represented by the PRS. Through ordinal logistic regression, we observed a beta coefficient of 0.808 and an associated odds ratio (OR) of 2.24 ($p = 1.65 \times 10^{-296}$). The remarkably small p -value emphasizes the robust statistical significance, signaling a substantial association between the PRS and familial T2D status. For each incremental unit rise in an individual’s PRS, their odds of belonging to a higher family

history category for T2D increase by 2.24 times. This implies a tangible shift in the likelihood of different family history classifications as the PRS changes. The findings underscore a strong statistical link between genetic predisposition, as captured by the PRS, and the gradation of family history of T2D.

Furthermore, we calculated the Population Attributable Risk (PAR) by dichotomizing PRS into a high-risk group (PRS tercile >80%) and a non-high-risk group (PRS tercile <80%). Among the 59,811 participants, the breakdown was as follows: high PRS with family history (N = 5,473), high PRS without family history (N = 6,489), non-high PRS with family history (N = 16,054), and non-high PRS without family history (N = 31,795). The PAR estimate was 10.17%, indicating that 10.17% of the family history of T2D is attributed to genetic heritability. If considering a broader definition of the high-risk group (PRS tercile >60%) and non-high-risk group (PRS tercile <60%), the PAR estimate increased to 18.41%.”

7. I don't really see the point of the on-line tool, it requires people to know their PRS and MRS. How would an individual have access to this information?

Response:

Thank you for the valuable feedback. Regarding this reviewer's comment on the on-line tool, we want to clarify that our website only requires age, sex, and family history as mandatory inputs. The use of PRS and MRS is optional and not mandatory for input. Therefore, individuals can utilize the tool with just age, sex, and family information to calculate their disease risk. We have revised the description in Lines 456 – 458 in the Results section in the revised manuscript as follows “We have established a website for users to calculate their T2D risk online. To obtain the risk assessment, users are required to provide age, sex, and family history of T2D and they can optionally provide PRS and MRS (Fig. 6F).”

It is important to emphasize that while PRS and MRS are not mandatory, they significantly contribute to a more accurate risk prediction. Users who have access to PRS and MRS data, which may be obtained through health check-ups, alternative means, or participation in related studies, can achieve enhanced precision in their risk assessment. For instance, the increasing availability of genetic and medical imaging data through health examinations has become more prevalent in recent years.

We value your insightful observation and will exercise caution in implementing and applying the tool. Should the reviewer persist in recommending the removal of our online risk calculator from this paper, we will follow the reviewer's suggestion.

8. Line 275 is confusing, it seems to suggest that HbA1c improves in T2D patients?

Response:

We rephrase the description in Lines 368 – 370 in the Results section in the revised manuscript as follows: “Additionally, the results also suggested that the T2D patients who regularly engaged in exercise can have a noteworthy improvement of 0.32% mg/dL in HbA1c than those T2D patients who did not do regular exercise.”

9. In line 425 the claim is too definitive, this could be resolved, e.g. by changing “revealed” to “suggested”.

Response:

Done. As recommended by this reviewer, we have rephrased the description as follows: “Further investigation suggests that these cases may represent individuals in a pre-T2D stage.” in Lines 548 – 549 of the Discussions section in the revised manuscript.

10. There seems to be a missing word in line 339 (“which MRS was not considered”)

Response:

Done. As recommended by this reviewer, we have revised the description as follows: “... the case vs. control ratio of sample size was 9.3 in the 90%–100% MRS decile group ...” Please refer to Lines 451 – 452 in the Discussions section in the revised manuscript.

References cited in this reply letter

- Auton, A., Brooks, L. D., Durbin, R. M., Garrison, E. P., Kang, H. M., Korbel, J. O., Marchini, J. L., McCarthy, S., McVean, G. A., & Abecasis, G. R. (2015). A global reference for human genetic variation. *Nature*, *526*(7571), 68-74. <https://doi.org/10.1038/nature15393> nature15393 [pii]
- Breiman, L., Friedman, J., Stone, C. J., & Olshen, R. A. (1984). *Classification and Regression Trees*. Taylor & Francis. <https://books.google.com.tw/books?id=JwQx-WOmSyQC>
- Brown, B. C., Ye, C. J., Price, A. L., & Zaitlen, N. (2016). Transethnic genetic-correlation estimates from summary statistics. *The American Journal of Human Genetics*, *99*(1), 76-88.
- Buchanan, G., John, J., Whiteside, A., Moisey, R., Malik, M., & Beer, S. (2009). Admission glucose is poor predictor of an abnormal glucose tolerance in acute coronary syndrome but abnormal oral glucose tolerance test predicts mortality. In: BMJ Publishing Group Ltd and British Cardiovascular Society.
- Ehehalt, S., Wiegand, S., Körner, A., Schweizer, R., Liesenkötter, K.-P., Partsch, C.-J., Blumenstock, G., Spielau, U., Denzer, C., & Ranke, M. B. (2017). Diabetes screening in overweight and obese children and adolescents: choosing the right test. *European journal of pediatrics*, *176*, 89-97.
- Fry, A., Littlejohns, T. J., Sudlow, C., Doherty, N., Adamska, L., Sprosen, T., Collins, R., & Allen, N. E. (2017). Comparison of sociodemographic and health-related characteristics of UK Biobank participants with those of the general population. *American journal of epidemiology*, *186*(9), 1026-1034.
- Ge, T., Chen, C.-Y., Ni, Y., Feng, Y.-C. A., & Smoller, J. W. (2019). Polygenic prediction via Bayesian regression and continuous shrinkage priors. *Nature Communications*, *10*(1), 1776. <https://doi.org/10.1038/s41467-019-09718-5>
- Greenhalgh, T., Campbell-Richards, D., Vijayaraghavan, S., Collard, A., Malik, F., Griffin, M., Morris, J., Claydon, A., & Macfarlane, F. (2011). New models of self-management education for minority ethnic groups: pilot randomized trial of a story-sharing intervention. *Journal of Health Services Research & Policy*, *16*(1), 28-36.
- John, R. M. (2022). The Well Pediatric Primary Care Visit and Screening Laboratory Tests. In *Pediatric Diagnostic Labs for Primary Care: An Evidence-based Approach* (pp. 101-134). Springer.
- Keyes, K. M., & Westreich, D. (2019). UK Biobank, big data, and the consequences of non-representativeness. *Lancet*, *393*(10178), 1297. [https://doi.org/10.1016/s0140-6736\(18\)33067-8](https://doi.org/10.1016/s0140-6736(18)33067-8)

- Li, G., Han, L., Wang, Y., Zhao, Y., Li, Y., Fu, J., Li, M., Gao, S., & Willi, S. M. (2018). Evaluation of ADA HbA1c criteria in the diagnosis of pre-diabetes and diabetes in a population of Chinese adolescents and young adults at high risk for diabetes: a cross-sectional study. *BMJ open*, *8*(8), e020665.
- Lin, Z., Guo, D., Chen, J., & Zheng, B. (2020). A nomogram for predicting 5-year incidence of type 2 diabetes in a Chinese population. *Endocrine*, *67*(3), 561-568. <https://doi.org/10.1007/s12020-019-02154-x>
- Mahajan, A., Spracklen, C. N., Zhang, W., Ng, M. C. Y., Petty, L. E., Kitajima, H., Yu, G. Z., Rueger, S., Speidel, L., Kim, Y. J., Horikoshi, M., Mercader, J. M., Taliun, D., Moon, S., Kwak, S. H., Robertson, N. R., Rayner, N. W., Loh, M., Kim, B. J., . . . Morris, A. P. (2022). Multi-ancestry genetic study of type 2 diabetes highlights the power of diverse populations for discovery and translation. *Nat Genet*, *54*(5), 560-572. <https://doi.org/10.1038/s41588-022-01058-3>
- Matafome, P. (2020). Epicardial adipose tissue (dys) function: A new player in heart disease? *Revista Portuguesa de Cardiologia*, *39*(11), 635-637.
- Nowicka, P., Santoro, N., Liu, H., Lartaud, D., Shaw, M. M., Goldberg, R., Guandalini, C., Savoye, M., Rose, P., & Caprio, S. (2011). Utility of hemoglobin A1c for diagnosing prediabetes and diabetes in obese children and adolescents. *Diabetes care*, *34*(6), 1306-1311.
- Panwar, H., Rashmi, H. M., Batish, V. K., & Grover, S. (2013). Probiotics as potential biotherapeutics in the management of type 2 diabetes—prospects and perspectives. *Diabetes/metabolism research and reviews*, *29*(2), 103-112.
- Ruan, Y., Lin, Y.-F., Feng, Y.-C. A., Chen, C.-Y., Lam, M., Guo, Z., Ahn, Y. M., Akiyama, K., Arai, M., Baek, J. H., Chen, W. J., Chung, Y.-C., Feng, G., Fujii, K., Glatt, S. J., Ha, K., Hattori, K., Higuchi, T., Hishimoto, A., . . . Stanley Global Asia, I. (2022). Improving polygenic prediction in ancestrally diverse populations. *Nature Genetics*, *54*(5), 573-580. <https://doi.org/10.1038/s41588-022-01054-7>
- Spiller, S., Blüher, M., & Hoffmann, R. (2018). Plasma levels of free fatty acids correlate with type 2 diabetes mellitus. *Diabetes, Obesity and Metabolism*, *20*(11), 2661-2669.
- Spracklen, C. N., Horikoshi, M., Kim, Y. J., Lin, K., Bragg, F., Moon, S., Suzuki, K., Tam, C. H. T., Tabara, Y., Kwak, S.-H., Takeuchi, F., Long, J., Lim, V. J. Y., Chai, J.-F., Chen, C.-H., Nakatochi, M., Yao, J., Choi, H. S., Iyengar, A. K., . . . Sim, X. (2020). Identification of type 2 diabetes loci in 433,540 East Asian individuals. *Nature*, *582*(7811), 240-245. <https://doi.org/10.1038/s41586-020-2263-3>
- Strings, S., Wells, C., Bell, C., & Tomiyama, A. (2023). The association of body mass index and odds of type 2 diabetes mellitus varies by race/ethnicity. *Public Health*, *215*, 27-30.

- Timothy Garvey, W. (2019). Clinical definition of overweight and obesity. *Bariatric Endocrinology: Evaluation and Management of Adiposity, Adiposopathy and Related Diseases*, 121-143.
- Wray, N. R., Goddard, M. E., & Visscher, P. M. (2008). Prediction of individual genetic risk of complex disease. *Current Opinion in Genetics & Development*, 18(3), 257-263. <https://doi.org/https://doi.org/10.1016/j.gde.2008.07.006>
- Yang, H.-C., Chu, S.-K., Huang, C.-L., Kuo, H.-W., Wang, S.-C., Liu, S.-W., Ho, I.-K., & Liu, Y.-L. (2016). Genome-Wide Pharmacogenomic Study on Methadone Maintenance Treatment Identifies SNP rs17180299 and Multiple Haplotypes on CYP2B6, SPON1, and GSG1L Associated with Plasma Concentrations of Methadone R- and S-enantiomers in Heroin-Dependent Patients. *PLOS Genetics*, 12(3), e1005910.

REVIEWERS' COMMENTS

Reviewer #1 (Remarks to the Author):

All comments have been addressed.

Reviewer #2 (Remarks to the Author):

The authors have adequately addressed my previous comments. Nevertheless, I still have several significant concerns. As previously noted, the accuracy of their prediction model, which incorporates polygenic risk scores (PRS), markedly improves when additional variables such as age, sex, PRS, and family history of the disease are included. When the model relies solely on PRS, its accuracy is notably inferior. The substantial impact of these additional variables on model accuracy remains unclear. This effect may be partially due to sampling bias, prompting my request for external validation.

Regrettably, the authors have not conducted the external validation as requested. Given the likelihood of sampling bias, I believe that the prediction model derived from their study, along with its accuracy estimates, may not be applicable to the general population. This limitation significantly undermines the utility and generalizability of their findings.

Reviewer #3 (Remarks to the Author):

Thank you for the detailed responses to my questions.

Authors' Reply

To Reviewer #1 (Remarks to the Author):

All comments have been addressed.

Re: We appreciate all the constructive comments by this reviewer.

Reviewer #2 (Remarks to the Author):

The authors have adequately addressed my previous comments. Nevertheless, I still have several significant concerns. As previously noted, the accuracy of their prediction model, which incorporates polygenic risk scores (PRS), markedly improves when additional variables such as age, sex, PRS, and family history of the disease are included. When the model relies solely on PRS, its accuracy is notably inferior. The substantial impact of these additional variables on model accuracy remains unclear. This effect may be partially due to sampling bias, prompting my request for external validation.

Regrettably, the authors have not conducted the external validation as requested. Given the likelihood of sampling bias, I believe that the prediction model derived from their study, along with its accuracy estimates, may not be applicable to the general population. This limitation significantly undermines the utility and generalizability of their findings.

Re: We appreciate this reviewer's suggestion for external validation and acknowledge the importance of further validation with an external dataset. We are dedicated to further validating our model and will actively work towards incorporating external datasets in subsequent studies.

Reviewer #3 (Remarks to the Author):

Thank you for the detailed responses to my questions.

Re: We appreciate all the constructive comments provided by this reviewer.